# Exact Random Graph Matching with Multiple Graphs

## Abstract

This work studies fundamental limits for recovering the underlying correspondence among *multiple* correlated random graphs. We identify a necessary condition for any algorithm to correctly match all nodes across all graphs, and propose two algorithms for which the same condition is also sufficient. The first algorithm employs global information to simultaneously match all the graphs, whereas the second algorithm first partially matches the graphs pairwise and then combines the partial matchings by transitivity. Both algorithms work down to the information theoretic threshold. Our analysis reveals a scenario where exact matching between two graphs alone is impossible, but leveraging more than two graphs allows exact matching among all the graphs. Along the way, we derive independent results about the $k$-core of Erdős-Rényi graphs.

## 1 Introduction

The information age has ushered an abundance of correlated networked data. For instance, the network structure of two social networks such as Facebook and Twitter is correlated because users are likely to connect with the same individuals in both networks. This wealth of correlated data presents both opportunities and challenges. On one hand, information from various datasets can be combined to increase the fidelity of data - translating to better performance in downstream learning tasks. On the other hand, the interconnected nature of this data also raises privacy and security concerns. Linkage attacks, for instance, exploit correlated data to identify individuals in an anonymized network by linking to other sources [NS09]. This poses a significant threat to user privacy.

Graph matching is the problem of recovering the underlying latent correspondence between correlated networks. The problem finds many applications in machine learning: de-anonymizing social networks [NS08, NS09], identifying similar functional components between species by matching their protein-protein interaction networks [BSI06, KHGPM16], object detection [SS05] and tracking [YYL+16] in computer vision, and textual inference for natural language processing [HNM05]. In most applications of interest, data is available in the form of *several* correlated networks. For instance, social media users are active each month on 6.7 social platforms on average [Ind23]. Similarly, reconciling protein-protein interaction networks among *multiple* species is an important problem in computational biology [SXB08]. As a first step toward this objective, many research works have studied the problem of matching *two* correlated graphs.

### 1.1 Related Work

The theoretical study of graph matching algorithms and their performance guarantees has primarily focused on Erdős-Rényi (ER) graphs. Pedarsani and Grossglauser [PG11] introduced the subsampling model to generate two such correlated graphs. The model entails twice subsampling each edge independently from a parent ER graph to obtain two sibling graphs, both of which are marginally ER graphs themselves. The goal is then to match nodes between the two graphs to recover the

underlying latent correspondence. This has been the framework of choice for many works that study graph matching. For example, Cullina and Kiyavash studied the problem of *exactly matching* two ER graphs, where the objective is to match *all* vertices correctly [CK16, CK17]. They identified a threshold phenomenon for this task: exact recovery is possible if the problem parameters are above a threshold, and impossible otherwise. Subsequently, threshold phenomena were also identified for *partial* graph matching between ER graphs - where the objective is to match only a positive fraction of nodes [GML21, HM23, WXY22, DD23]. The case of almost-exact recovery - where the objective is to match all but a negligible fraction of nodes - was studied by Cullina and co-authors: a necessary condition for almost exact recovery was identified, and it was shown that the same condition is also sufficient for the *k-core estimator* [CKMP19]; the estimator is described formally in Section 3. This estimator proved useful to uncover the fundamental limits for graph matching in other contexts such as the stochastic block model [GRS22] and inhomogeneous random graphs [RS23]. Ameen and Hajek [AH23] showed some robustness properties of the $k$-core estimator in the context of matching ER graphs under node corruption. The estimator plays an important role in the present work as well.

A sound understanding of ER graphs inspires algorithms for real-world networks. Various *efficient* algorithms have been proposed, including algorithms based on the spectrum of the graph adjacency matrices [FMWX22], node degree and neighborhood based algorithms [DCKG19, DMWX21, MRT23] as well as algorithms based on iterative methods [DL23] and counting subgraphs [MWXY23, BCL$^+$19]. Some of these are discussed in Section 5 in relation to the present work.

Incorporating information from multiple graphs to match them has been recognized as an important research direction, for instance in the work of Gaudio and co-authors [GRS22]. To our knowledge, the only other papers to consider matchings among multiple graphs are the works of Josephs and co-authors [JLK21], and of Rácz and Sridhar [RS21]. However, these works have different objectives and are not concerned with the fundamental limits for matching $m$ graphs. In fact, both works note that it is possible to exactly match $m$ graphs whenever it is possible to exactly match any two graphs by pairwise matching all the graphs exactly. In contrast, we show that under appropriate conditions, it is possible to exactly match $m$ ER graphs even when no two graphs can be pairwise matched exactly.

**Contributions** In this work, we investigate the problem of combining information from *multiple* correlated networks to boost the number of nodes that are correctly matched among them. We consider the natural generalization of the subsampling model to generate $m$ correlated random graphs, and identify a threshold such that it is impossible for any algorithm to match all nodes correctly across all graphs when the problem parameters are below this threshold. Conversely, we show that exact recovery is possible above the threshold. This characterization generalizes known results for exact graph matching when $m = 2$. Subsequently, we show that there is a region in parameter space for which exactly matching any two graphs is impossible using only the two graphs, and yet exact graph matching is possible among $m > 2$ graphs using all the graphs.

We present two algorithms and prove their optimality for this task. The first algorithm matches all $m$ graphs simultaneously based on global information about the graphs. In contrast, the second algorithm first *pairwise* matches graphs, and then combines them to match all nodes across all graphs. We show that both algorithms correctly match all the graphs all the way down to the information theoretic threshold. Finally, we illustrate through simulation that our subroutine to combine information from pairwise comparisons between networks works well when paired with efficient algorithms for graph matching. Our analysis also yields some theoretical results about the $k$-core of ER graphs that are of independent interest.

# 2 Preliminaries and Setup

**Notation** In this work, $G \sim \mathsf{ER}(n, p)$ denotes that the graph $G$ is sampled from the Erdős-Rényi distribution with parameters $n$ and $p$, i.e. $G$ has $n$ nodes and each edge is independently present with probability $p$. For a graph $G$, we denote the set of its vertices by $V \equiv V(G)$ and its edges by $E(G)$. The *edge status* of each vertex pair $\{i, j\}$ with $i \neq j$ is denoted by $G\{i.j\}$, so that $G\{i, j\} = 1$ if $\{i, j\} \in E(G)$ and $G\{i, j\} = 0$ otherwise. The degree of a node $v$ in graph $G$ is denoted $\delta_G(v)$. Let $\pi$ denote a permutation on $V(G) = \{1, \cdots, n\}$. For a graph $G$, denote by $G^\pi$ the graph obtained by permuting the nodes of $G$ according to $\pi$, so that

$$G\{i, j\} = G^\pi \{\pi(i), \pi(j)\} \ \forall \, i, j \in V(G) \text{ such that } i \neq j.$$

Standard asymptotic notation $(O(\cdot), o(\cdot), \cdots)$ is used throughout and it is implicit that $n \to \infty$.

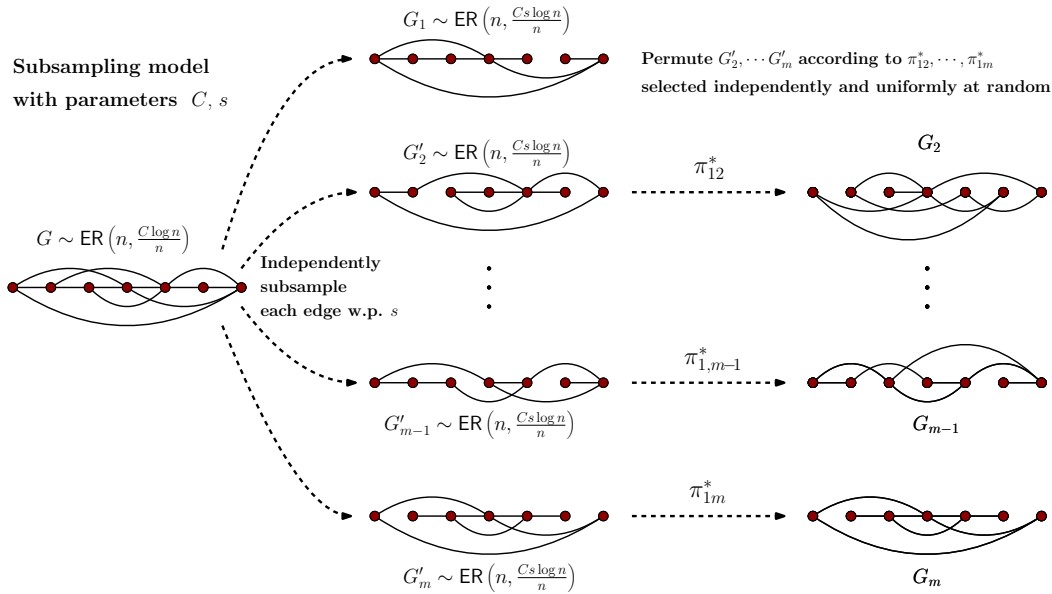

Figure 1: Illustration of obtaining $m$ correlated graphs from the subsampling model

**Subsampling model**   Consider the subsampling model for correlated random graphs [PG11], which has a natural generalization to the setting of $m$ graphs. In this model, a parent graph $G$ is sampled from the Erdős-Rényi distribution $\mathsf{ER}(n, p)$. The $m$ graphs $G_1, G'_2, \cdots, G'_{m-1}, G'_m$ are obtained by independently subsampling each edge from $G$ with probability $s$. Finally, the graphs $G_2, \cdots, G_m$ are obtained by permuting the nodes of each of the graphs $G'_2, \cdots, G'_m$ respectively according to independent permutations $\pi^*_{12}, \cdots, \pi^*_{1m}$ sampled uniformly at random from the set of all permutations on $[n]$, i.e.

$$G_j = (G'_j)^{\pi^*_{1j}} \text{ for all } j \in \{2, \cdots, m\}.$$

Figure 1 illustrates this process of obtaining correlated graphs using the subsampling model. In this work, we are interested in the setting where $s$ is constant and $p = C \log(n)/n$ for some $C > 0$.

**Objective 1.** *Determine conditions on parameters $C$, $s$ and $m$ so that given correlated graphs $G_1, \cdots, G_m$ from the subsampling model, it is possible to exactly recover the underlying correspondences $\pi^*_{12}, \cdots, \pi^*_{1m}$ with probability $1 - o(1)$.*

Stated thus, the underlying correspondences use the graph $G_1$ as a reference. Thus, for ease of notation, we will use $G_1$ and $G'_1$ interchangeably. Note that the underlying correspondence between all the graphs is fixed upon fixing $\pi^*_{12}, \cdots, \pi^*_{1m}$: for any two graphs $G_i$ and $G_j$, their underlying correspondence is given by $\pi^*_{ij} := \pi^*_{1j} \circ (\pi^*_{1i})^{-1}$.

Formally, a *matching* $(\mu_{12}, \cdots, \mu_{1m})$ is a collection of injective functions with domain $\mathrm{dom}(\mu_{1i}) \subseteq V$ for each $i$, and co-domain $V$. An *estimator* is simply a mechanism to map any collection of graphs $(G_1, \cdots, G_m)$ to a matching. We say that an estimator *completely* matches the graphs if the output mappings $\mu_{12}, \cdots \mu_{1m}$ are all complete, i.e. they are all permutations on $\{1, \cdots, n\}$.

# 3   Main Results and Algorithm

This section presents necessary and sufficient conditions to meet Objective 1.

**Theorem 2** (Impossibility). *Let $G_1, \cdots, G_m$ be correlated graphs obtained from the subsampling model with parameters $C$ and $s$, and let $\pi^*_{12}, \cdots, \pi^*_{1m}$ denote the underlying latent correspondences between $G_1$ and $G_2, \cdots, G_m$ respectively. Suppose that*

$$Cs \left(1 - (1-s)^{m-1}\right) < 1.$$

*The output $\widehat{\pi}_{12}, \cdots, \widehat{\pi}_{1m}$ of any estimator satisfies*

$$\mathbb{P}\left(\widehat{\pi}_{12} = \pi^*_{12},\ \widehat{\pi}_{13} = \pi^*_{13}, \cdots,\ \widehat{\pi}_{1m} = \pi^*_{1m}\right) = o(1).$$

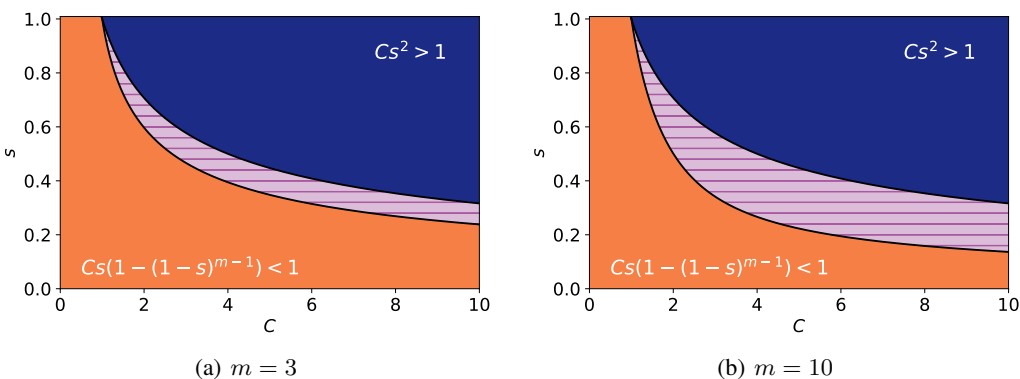

(a) $m = 3$             (b) $m = 10$

Figure 2: Regions in parameter space. *Orange*: Exactly matching $m$ graphs is impossible even with $m$ graphs. *Blue*: Exactly matching 2 graphs is possible with 2 graphs. *Striped*: Impossible to match 2 graphs using only the 2 graphs, but possible using $m$ graphs as side information.

Theorem 2 implies that the condition $Cs(1 - (1-s)^{m-1} > 1$ is a necessary condition to exactly match $m$ graphs with probability bounded away from 0. We show that this condition is also sufficient to exactly match $m$ graphs with probability going to 1.

**Theorem 3** (Achievability). *Let $G_1, \cdots, G_m$ be correlated graphs obtained from the subsampling model with parameters $C$ and $s$, and let $\pi_{12}^*, \cdots, \pi_{1m}^*$ denote the underlying latent correspondences between $G_1$ and $G_2, \cdots, G_m$ respectively. Suppose that*

$$Cs\left(1 - (1-s)^{m-1}\right) > 1.$$

*There is an estimator whose output $\widehat{\pi}_{12}, \cdots, \widehat{\pi}_{1m}$ satisfies*

$$\mathbb{P}\left(\widehat{\pi}_{12} = \pi_{12}^*,\ \widehat{\pi}_{13} = \pi_{13}^*, \cdots,\ \widehat{\pi}_{1m} = \pi_{1m}^*\right) = 1 - o(1).$$

Theorems 2 and 3 together characterize the threshold for exact recovery. A few remarks are in order.

1. For $m = 2$, the condition $Cs(1 - (1-s)^{m-1}) > 1$ reduces to $Cs^2 > 1$, which is known to be necessary and sufficient for exactly matching two graphs [CK17, WXY22].

2. For any $m > 2$, there is a non-empty region in the parameter space defined by

$$Cs(1 - (1-s)^{m-1}) > 1 > Cs^2.$$

    For any $C$ and $s$ in this region, it is impossible to exactly match any two graphs $G_i$ and $G_j$ without using the other $m - 2$ graphs as side information. Upon using them, however, it is possible to exactly match all nodes across the $m$ graphs. This is illustrated in Figure 2.

## 3.1 Algorithms for exact recovery

For any two graphs $H_1$ and $H_2$ on the same vertex set $V$, denote by $H_1 \vee H_2$ their *union graph* and by $H_1 \wedge H_2$ their *intersection graph*. An edge $\{i, j\}$ is present in $H_1 \vee H_2$ if it is present in either $H_1$ or $H_2$. Similarly, the edge is present in $H_1 \wedge H_2$ if it is present in both $H_1$ and $H_2$.

A natural starting point is to study the maximum likelihood estimator (MLE) because it is optimal. To that end, we compute the log-likelihood function; the details are deferred to Appendix A.

**Theorem 4.** *Let $\pi_{12}, \cdots, \pi_{1m}$ denote a collection of permutations on $\{1, \cdots, n\}$. Then*

$$\log \mathbb{P}\left(G_1, \cdots, G_m \mid \pi_{12}^* = \pi_{12}, \cdots, \pi_{1m}^* = \pi_{1m}\right) \propto const. - \left|E\left(G_1 \vee G_2^{\pi_{12}} \vee \cdots \vee G_m^{\pi_{1m}}\right)\right|,$$

*where const. depends only on $p$, $s$ and $G_1, \cdots, G_m$.*

Theorem 4 reveals that the MLE for exactly matching $m$ graphs has a neat interpretation: simply pick $\pi_{12}, \cdots, \pi_{1m}$ to minimize the number of edges in the corresponding union graph. This is presented as Algorithm 1. Despite this nice interpretation of the MLE, its analysis is quite cumbersome. We instead present and analyze a different estimator, presented as Algorithm 2.

---

**Algorithm 1:** Maximum likelihood estimator

---

**require:** Graphs $G_1, G_2, \cdots, G_m$ on a common vertex set $V$

**1 for** $(\pi_{12}, \pi_{13}, \cdots, \pi_{1m})$ *such that each $\pi_{1j}$ is a permutation on $[n]$* **do**

**2** $\quad\big|\quad W(\pi_{12}, \cdots, \pi_{1m}) \leftarrow |E(G_1 \vee G_2^{\pi_{12}} \vee \cdots \vee G_m^{\pi_{1m}})|$

**3 end**

**4 return** $(\widehat{\pi}_{12}^{\mathrm{ML}}, \cdots, \widehat{\pi}_{1m}^{\mathrm{ML}}) \in \arg\max_{\pi_{12}, \cdots, \pi_{1m}} W(\pi_{12}, \cdots, \pi_{1m})$

---

---

**Algorithm 2:** Matching through transitive closure

---

**require:** Graphs $G_1, G_2, \cdots, G_m$ on a common vertex set $V$, Integer $k$

`// Step 1: Pairwise matching`

**1 for** $\{i, j\}$ *in* $\{1, \cdots, m\}$ *such that $i < j$* **do**

**2** $\quad\big|\quad \widehat{\nu}_{ij} \leftarrow \arg\max_\pi |\mathsf{core}_k (G_i \wedge G_j^\pi)|$

**3** $\quad\big|\quad \widehat{\mu}_{ij} \leftarrow \widehat{\nu}_{ij}$ with domain restricted to $\mathsf{core}_k(G_i \wedge G_j^{\widehat{\nu}_{ij}})$ `         // k-core estimator`

**4 end**

`// Step 2: Boosting through transitive closure`

**5 for** $v \in V$ **do**

**6** $\quad\big|\quad$ **for** $j = 2, \cdots, m$ **do**

**7** $\quad\big|\quad\big|\quad$ **if** *there is a sequence of indices $1 = k_1, \cdots, k_\ell = j$ in $[m]$ such that* $\widehat{\mu}_{k_{\ell-1},j} \circ \cdots \circ \widehat{\mu}_{k_2,k_3} \circ \widehat{\mu}_{1,k_2}(v) = v'$ *for some $v' \in [n]$* **then**

**8** $\quad\big|\quad\big|\quad\big|\quad$ Set $\widehat{\pi}_{1j}(v) = v'$

**9** $\quad\big|\quad\big|\quad$ **end**

**10** $\quad\big|\quad$ **end**

**11 end**

**12 return** $\widehat{\pi}_{12}, \cdots, \widehat{\pi}_{1m}$

---

Algorithm 2 runs in two steps: In step 1, the $k$-core estimator, for a suitable choice of $k$, is used to pairwise match all the graphs. For any $i$ and $j$, the $k$-core estimator selects a permutation $\widehat{\nu}_{ij}$ to maximize the size of the $k$-core[1] of $G_i \wedge G_j^{\widehat{\nu}_{ij}}$. It then outputs a matching $\widehat{\mu}_{ij}$ by restricting the domain of $\widehat{\nu}_{ij}$ to $\mathsf{core}_k(G_i \wedge G_j^{\widehat{\nu}_{ij}})$. These matchings $\widehat{\mu}_{ij}$ need not be complete - in fact, each of them is a partial matching with high probability whenever $Cs^2 < 1$. In step 2, these partial matchings are *boosted* as follows: If a node $v$ is unmatched between two graphs $G_i$ and $G_j$, then search for a sequence of graphs $G_i, G_{k_1}, \cdots, G_{k_\ell}, G_j$ such that $v$ is matched between any two consecutive graphs in the sequence. If such a sequence exists, then extend $\widehat{\mu}_{i,j}$ to include $v$ by transitively matching it from $G_i$ to $G_j$.

In Section 4.2, we show that Algorithm 2 correctly matches all nodes across all graphs with probability $1 - o(1)$, whenever the necessary condition $Cs(1 - (1-s)^{m-1}) > 1$ holds. We remark that this also implies that Algorithm 1 succeeds under the same condition, because the MLE is optimal. Note that the MLE selects all permutations $\widehat{\pi}_{12}, \cdots, \widehat{\pi}_{1m}$ *simultaneously* based on their union graph. In contrast, Algorithm 2 only ever makes *pairwise* comparisons between graphs. Perhaps surprisingly, it turns out that this is sufficient for exact recovery. An analysis of Algorithm 2 is presented in Section 4. Along the way, independent results of interest on the $k$-core of Erdős-Rényi graphs are obtained.

---

[1]The $k$-core of a graph $G$ is the largest subset of vertices $\mathsf{core}_k(G)$ such that the induced subgraph has minimum degree at least $k$.

## 4  Proof Outlines and Key Insights

### 4.1  Impossibility of exact graph matching (Theorem 2)

This result has a simple proof following a genie-aided converse argument. The idea is to reduce the problem to that of matching two graphs by providing extra information to the estimator.

*Proof of Theorem 2.* If the correspondences $\pi_{12}^*, \cdots, \pi_{1,m-1}^*$ were provided as extra information to an estimator, then the estimator must still match $G_m$ with the union graph $G_1' \vee G_2' \vee \cdots \vee G_{m-1}'$. This can be viewed as an instance of matching two graphs obtained by *asymmetric* subsampling: the graph $G_m$ is obtained from a parent graph $G \sim \mathsf{ER}(n, C\log(n)/n)$ by subsampling each edge independently with probability $s_1 := s$, and the graph $\widetilde{G}_{m-1} := G_1' \vee G_2' \vee \cdots \vee G_{m-1}'$ is obtained from $G$ by subsampling each edge independently with probability $s_2 := 1 - (1-s)^{m-1}$. Cullina and Kiyavash studied this model for matching two graphs: Theorem 2 of [CK17] establishes that matching $G_m$ and $\widetilde{G}_{m-1}$ is impossible if $Cs_1 s_2 < 1$, or equivalently if $Cs(1 - (1-s)^{m-1}) < 1$. $\square$

### 4.2  Achievability of exact graph matching (Theorem 3)

Algorithm 2 succeeds if both step 1 and step 2 succeed, i.e.

1. Each instance of pairwise matching using the $k$-core estimator is correct on its domain, i.e.
$$\widehat{\mu}_{ij}(v) = \pi_{ij}^*(v) \; \forall v \in \mathsf{dom}(\widehat{\mu}_{ij}), \; \forall i, j.$$

2. For each node $v$ and any two graphs $G_i$ and $G_j$, there is a sequence of graphs such that $v$ can be transitively matched through those graphs between $G_i$ and $G_j$.

**On step 1**  This falls back to the regime of analyzing the performance of the $k$-core estimator in the setting of two graphs. Cullina and co-authors [CKMP19] showed that the $k$-core estimator is *precise*: For any two correlated graphs $G_i$ and $G_j$ with $p = C\log(n)/n$ and constant $s$, the $k$-core estimator correctly matches all nodes in $\mathsf{core}_k(G_i' \wedge G_j')$ with probability $1 - o(1)$. In fact, this is true for any $C > 0$ and for any $k \geq 13$ [RS23]. Therefore, using the fact that the number of instances of pairwise matchings is constant whenever $m$ is constant, a union bound reveals

$$\mathbb{P}(\exists \, 1 \leq i < j \leq m \text{ such that } \widehat{\mu}_{ij}(v) \neq \pi_{ij}^*(v) \text{ for some } v \in \mathsf{core}_k(G_i' \wedge G_j'))$$
$$\leq \sum_{i=1}^m \sum_{j=1}^m \mathbb{P}\left(\widehat{\mu}_{i,j}(v) \neq \pi_{i,j}^*(v) \text{ for some } v \in \mathsf{core}_k(G_i' \wedge G_j')\right)$$
$$= o(1).$$

We have proved the following.

**Proposition 5.** *Let $G_1, \cdots, G_m$ be correlated graphs from the subsampling model. Let $k \geq 13$ and let $\widehat{\mu}_{ij}$ denote the matching output by the $k$-core estimator on graphs $G_i$ and $G_j$. Then,*
$$\mathbb{P}(\exists \, 1 \leq i < j \leq m, \text{ and } v \in \mathsf{core}_k(G_i' \wedge G_j')) \text{ such that } \widehat{\mu}_{ij}(v) \neq \pi_{ij}^*(v)) = o(1).$$

**On step 2**  The challenging part of the proof is to show that boosting through transitive closure matches all the nodes with probability $1 - o(1)$ if $Cs(1 - (1-s)^{m-1}) > 1$. It is instructive to visualize this using *transitivity graphs*.

**Definition 6** (Transitivity graph, $\mathcal{H}(v)$)**.** *For each node $v \in V$, let $\mathcal{H}(v)$ denote the graph on the vertex set $\{g_1, \cdots, g_m\}$ such that an edge $\{g_i, g_j\}$ is present in $\mathcal{H}(v)$ if and only if $v \in \mathsf{core}_k(G_i' \wedge G_j')$.*

On the event that each instance of pairwise matching using the $k$-core is correct, the edge $\{g_i, g_j\}$ is present in $\mathcal{H}(v)$ if and only if $v$ is correctly matched using the $k$-core estimator between $G_i$ and $G_j$, i.e. $\pi_{1i}^*(v)$ is matched to $\pi_{1j}^*(v)$. Thus, in order for Step 2 to succeed (i.e. to exactly match all vertices across all graphs), it suffices that the graph $\mathcal{H}(v)$ is connected for each node $v \in V$. However, studying the connectivity of the transitivity graphs is challenging because in any graph $\mathcal{H}(v)$, no two edges are independent. This is because the $k$-cores of any two intersection graphs $G_a' \wedge G_b'$ and $G_c' \wedge G_d'$ are correlated, because all the graphs $G_a, G_b, G_c$ and $G_d$ are themselves correlated. To overcome this, we introduce another graph $\widetilde{\mathcal{H}}(v)$ that relates to $\mathcal{H}(v)$ and is amenable to analysis.

**Definition 7.** *For each node $v \in V$, let $\widetilde{\mathcal{H}}(v)$ denote a complete weighted graph on the vertex set $\{g_1, \cdots, g_m\}$ such that the weight on any edge $\{g_i, g_j\}$ is $\widetilde{c}_v(i,j) := \delta_{G_i' \wedge G_j'}(v)$.*

The relationship between the graphs $\mathcal{H}(v)$ and $\widetilde{\mathcal{H}}(v)$ stems from a useful relationship between the degree of node $v$ in $G_i' \wedge G_j'$ and the inclusion of $v$ in $\mathsf{core}_k(G_i' \wedge G_j')$ for each $i$ and $j$. Since this result is of independent interest in the study of random graphs, we state it below for general Erdős-Rényi graphs.

**Lemma 8.** *Let $n$ and $k$ be positive integers and let $G \sim \mathsf{ER}(n, \alpha \log(n)/n)$ for some $\alpha > 0$. Let $v$ be a node of $G$ and let $\delta_G(v)$ denote the degree of $v$ in $G$. Then,*

$$\mathbb{P}\left(\{v \notin \mathsf{core}_k(G)\} \cap \{\delta_G(v) \geq k + 1/\alpha\}\right) = o\left(1/n\right). \tag{1}$$

For any $i$ and $j$, the graph $G_i' \wedge G_j' \sim \mathsf{ER}(n, Cs^2 \log(n)/n)$. Thus, Lemma 8 implies that with probability $1 - o(1/n)$, if a pair $\{g_i, g_j\}$ has edge weight $\widetilde{c}_{ij} \geq k + 1/\alpha$ in $\widetilde{\mathcal{H}}(v)$, then the corresponding edge $\{g_i, g_j\}$ is present in the transitivity graph $\mathcal{H}(v)$. Equivalently, $v$ is correctly matched between $G_i$ and $G_j$ in the instance of pairwise $k$-core matching between them.

The graph $\mathcal{H}(v)$ is not connected only if it contains a (non-empty) vertex cut $U \subset \{1, \cdots, m\}$ with no edge crossing between $U$ and $U^c$. Let $c_v(U)$ denote the number of such crossing edges in $\mathcal{H}(v)$. Furthermore, define the *cost* of the cut $U$ in $\widetilde{\mathcal{H}}(v)$ as

$$\widetilde{c}_v(U) := \sum_{i \in U} \sum_{j \in U^c} \widetilde{c}_v(i,j).$$

Lemma 8 is a statement about a single graph, but we show it can be invoked to prove the following.

**Theorem 9.** *Let $G_1, \cdots, G_m$ be correlated graphs from the subsampling model with parameters $C$ and $s$. Let $v \in V$ and let $U$ be a vertex cut of $\{1, \cdots, m\}$ such that $|U| \leq \lfloor m/2 \rfloor$. Then,*

$$\mathbb{P}\left(\{c_v(U) = 0\} \cap \left\{\widetilde{c}_v(U) > \frac{m^2}{4}\left(k + \frac{1}{Cs^2}\right)\right\}\right) = o(1/n). \tag{2}$$

It suffices therefore to analyze the probability that the graph $\widetilde{\mathcal{H}}(v)$ has a cut $U$ such that its cost $\widetilde{c}_v(U)$ is too small. To that end, we show that the bottleneck arises from vertex cuts of small size. Formally,

**Theorem 10.** *Let $G_1, \cdots, G_m$ be correlated graphs from the subsampling model. Let $v \in V$ and let $U_\ell$ denote the set $\{1, \cdots, \ell\}$ for $\ell$ in $\{1, \cdots, \lfloor m/2 \rfloor\}$. For any vertex cut $U$ of $\{1, \cdots, m\}$, let $\widetilde{c}_v(U)$ denote its cost in the graph $\widetilde{\mathcal{H}}(v)$. The following stochastic ordering holds:*

$$\widetilde{c}_v(U_1) \preceq \widetilde{c}_v(U_2) \preceq \cdots \preceq \widetilde{c}_v(U_{\lfloor m/2 \rfloor}).$$

Theorems 9 and 10 imply that the tightest bottleneck to the connectivity of $\mathcal{H}(v)$ is the event that $\widetilde{c}_v(U_1)$ is below the threshold $r := \frac{m^2}{4}\left(k + \frac{1}{Cs^2}\right)$, i.e. the sum of degrees of $v$ over the intersection graphs $(G_1 \wedge G_j' : j = 2, \cdots, m)$ is less than $r$. This event occurs only if the degree of $v$ is less than $r$ in each of the intersection graphs $(G_1 \wedge G_j' : j = 2, \cdots, m)$. However, under the condition $Cs(1 - (1-s)^{m-1}) > 1$, it turns out that this event occurs with probability $o(1/n)$.

**Theorem 11.** *Let $G_1, \cdots, G_m$ be obtained from the subsampling model with parameters $C$ and $s$. Let $r = \frac{m^2}{4}\left(k + \frac{1}{Cs^2}\right)$. Let $v \in [n]$ and suppose that $Cs(1 - (1-s)^{m-1}) > 1$. Then,*

$$\mathbb{P}\left(\widetilde{c}_v(U_1) \leq r\right) \leq \mathbb{P}\left(\left\{\delta_{G_1 \wedge G_2'}(v) \leq r\right\} \cap \cdots \cap \left\{\delta_{G_1 \wedge G_m'}(v) \leq r\right\}\right) = o\left(1/n\right).$$

### 4.3 Piecing it all together: Proof of Theorem 3

*Proof of Theorem 3.* Let $\widehat{\pi}_{12}, \cdots, \widehat{\pi}_{1m}$ denote the output of Algorithm 2 with $k \geq 13$. Let $E_1$ (resp. $E_2$) denote the event that Algorithm 1 (resp. Algorithm 2) fails to match all $m$ graphs exactly, i.e.

$$E_1 = \left\{\widehat{\pi}_{12}^{\mathsf{ML}} \neq \pi_{12}^*\right\} \cup \cdots \cup \left\{\widehat{\pi}_{1m}^{\mathsf{ML}} \neq \pi_{1m}^*\right\}, \qquad E_2 = \left\{\widehat{\pi}_{12} \neq \pi_{12}^*\right\} \cup \cdots \cup \left\{\widehat{\pi}_{1m} \neq \pi_{1m}^*\right\}.$$

First, we show that the output of Algorithm 2 is correct with probability $1 - o(1)$ whenever $Cs(1 - (1-s)^{m-1}) > 1$. If the event $E_2$ occurs, then either step 1 failed, i.e. there is a $k$-core matching $\widehat{\mu}_{ij}$ that is incorrect, or step 2 failed, i.e. at least one of the graphs $\mathcal{H}(v)$ is not connected. Therefore,

$$\mathbb{P}\left(E_2\right) \leq \mathbb{P}\left(\bigcup_{i,j} \bigcup_{v \in \mathsf{core}_k(G_i' \wedge G_j')} \{\widehat{\mu}_{ij} \neq \pi_{ij}^*\}\right) + \mathbb{P}\left(\bigcup_{v \in V} \{\mathcal{H}(v) \text{ is not connected}\}\right) \leq o(1) + \sum_{v \in V} q_v,$$

where the last step uses Proposition 5, and $q_v$ denotes the probability that the transitivity graph $\mathcal{H}(v)$ is not connected. For each $\ell$ in the set $\{1, \cdots, \lfloor m/2 \rfloor\}$, let $U_\ell$ denote the set $\{1, \cdots, \ell\}$. Then,

$$q_v = \mathbb{P}\left(\bigcup_{\ell=1}^{\lfloor m/2 \rfloor} \{\exists\, U \subset \{1, \cdots, m\} : |U| = \ell \text{ and } c_v(U) = 0\}\right)$$

$$\leq \sum_{\ell=1}^{\lfloor m/2 \rfloor} \binom{m}{\ell} \cdot \mathbb{P}\left(c_v(U_\ell) = 0\right)$$

$$\leq \sum_{\ell=1}^{\lfloor m/2 \rfloor} \binom{m}{\ell} \left[\mathbb{P}\left(\widetilde{c}_v(U_\ell) \leq \frac{m^2}{4}\left(k + \frac{1}{Cs^2}\right)\right) + \mathbb{P}\left(\{c_v(U_\ell) = 0\} \cap \left\{\widetilde{c}_v(U_\ell) > \frac{m^2}{4}\left(k + \frac{1}{Cs^2}\right)\right\}\right)\right]$$

$$\overset{(a)}{\leq} \sum_{\ell=1}^{\lfloor m/2 \rfloor} \binom{m}{\ell} \left[\mathbb{P}\left(\widetilde{c}_v(U_\ell) \leq \frac{m^2}{4}\left(k + \frac{1}{Cs^2}\right)\right) + o\left(\frac{1}{n}\right)\right]$$

$$\overset{(b)}{\leq} \sum_{\ell=1}^{\lfloor m/2 \rfloor} \binom{m}{\ell} \left[\mathbb{P}\left(\widetilde{c}_v(U_1) \leq \frac{m^2}{4}\left(k + \frac{1}{Cs^2}\right)\right) + o\left(\frac{1}{n}\right)\right]$$

$$\overset{(c)}{\leq} \sum_{\ell=1}^{\lfloor m/2 \rfloor} m^\ell \left[o\left(\frac{1}{n}\right) + o\left(\frac{1}{n}\right)\right] = o\left(\frac{1}{n}\right).$$

Here, (a) uses Theorem 9, and (b) uses the fact that for any $\ell \geq 2$, the random variable $\widetilde{c}_v(U_\ell)$ stochastically dominates $\widetilde{c}_v(U_1)$ (Theorem 10). Finally, (c) uses Theorem 11 and the fact that $Cs(1 - (1-s)^{m-1}) > 1$. Therefore, a union bound over all the nodes yields

$$\mathbb{P}\left(E_2\right) \leq o(1) + \sum_{v \in V} q_v \leq o(1) + n \times o(1/n) = o(1).$$

Finally, by optimality of the MLE, it follows that

$$\mathbb{P}\left(E_1\right) \leq \mathbb{P}\left(E_2\right) = o(1),$$

whenever $Cs(1 - (1-s)^{m-1}) > 1$. This concludes the proof. $\qquad \square$

# 5   Discussion and Future Work

In this work, we introduced and analyzed matching through transitive closure - an approach that combines information from multiple graphs to recover the underlying correspondence between them. Despite its simplicity, it turns out that matching through transitive closure is an optimal way to combine information in the setting where the graphs are pairwise matched using the $k$-core estimator. A limitation of our algorithms is the runtime: Algorithm 2 does not run in polynomial time because it uses the $k$-core estimator for pairwise matching, which involves searching over the space of permutations. Even so, it is useful to establish the fundamental limits of exact recovery, and serve as a benchmark to compare the performance of any other algorithm.

The transitive closure subroutine (Step 2) itself is *efficient* because it runs in polynomial time $O(mn)$. Therefore, a natural next step is to modify Step 1 in our algorithm so that the pairwise matchings are done by an *efficient* algorithm. However, it is not clear if transitive closure is optimal for combining information from the pairwise matchings in this setting. For example, there is a possibility that the pairwise matchings resulting from the efficient algorithm are heavily correlated, and transitive closure is unable to boost them. In Figure 3, we show experimentally that this is not the case for two algorithms of interest: GRAMPA [FMWX22] and Degree Profiles [DMWX21].

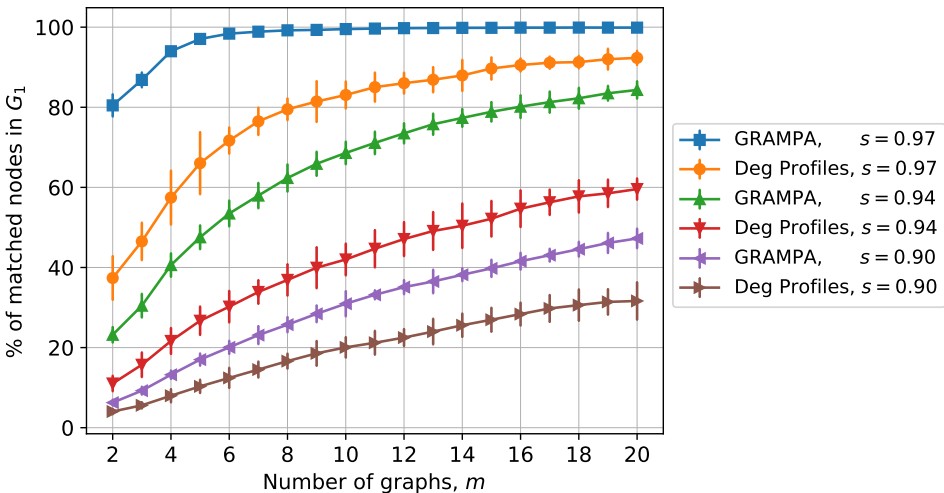

Figure 3: Matching through transitive closure

1. `GRAMPA` is a spectral algorithm that uses the entire spectrum of the adjacency matrices to match the two graphs. The code is available in [FMWX20].

2. `Degree Profiles` associates with each node a signature derived from the degrees of its neighbors, and matches nodes by signature proximity. The code is available in [DMWX20].

Evidently, both algorithms benefit substantially from using transitive closure to boost the number of matched nodes. This suggests that transitive closure can be a practical algorithm to boost matchings between networks by using other networks as side-information. Unfortunately, both `GRAMPA` and `Degree Profiles` require the graphs to be close to isomorphic in order to perform well, and so they do not perform well when the model parameters are close to the information theoretic threshold for exact recovery. Subsequently, they cannot be used to answer the question in Objective 1.

Our work presents several directions for future research.

- **Polynomial-time algorithms.** Using a polynomial-time estimator in place of the $k$-core estimator in Step 1 of Algorithm 2 yields a polynomial-time algorithm to match $m$ graphs. It is critical that the estimator in question is able to identify for itself the nodes that it has matched correctly - this precision is present in the $k$-core estimator and enables the transitive closure subroutine to work correctly. Can the performance guarantees of the $k$-core estimator be realized through polynomial time algorithms that meet this constraint?

- **Beyond Erdős-Rényi graphs.** The study of matching *two* ER graphs provided tools and techniques that extended to the analysis of more realistic models. For instance, the $k$-core estimator itself played a crucial role in establishing limits to matching two correlated stochastic block models [GRS22] and two inhomogeneous random graphs [RS23]. Can the techniques developed in the present work be used to identify the information theoretic limits to exact recovery in these models in the general setting of $m$ graphs?

- **Boosting for partial recovery.** This work focused on *exact* recovery, where the objective is to match *all* nodes across *all* graphs. It would be interesting to consider a regime where any instance of pairwise matching recovers at best a small fraction of nodes. Is it possible to quantify the extent to which transitive closure boosts the number of matched nodes?

- **Robustness.** Finally, how sensitive to perturbation is the transitive closure algorithm? Is it possible to quantify the extent to which an adversary may perturb edges in some of the graphs without losing the performance guarantees of the matching algorithm? Algorithms that perform well on models such as ER graphs and are further generally robust are expected to also work well with real-world networks.

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

## A Maximum Likelihood Estimator

Recall the form of the maximum likelihood estimator as claimed in Theorem 4.

**Theorem 4.** *Let $\pi_{12}, \cdots, \pi_{1m}$ denote a collection of permutations on $\{1, \cdots, n\}$. Then*

$$\log \mathbb{P}\left(G_1, \cdots, G_m \mid \pi_{12}^* = \pi_{12}, \cdots, \pi_{1m}^* = \pi_{1m}\right) \propto \text{const.} - \left|E\left(G_1 \vee G_2^{\pi_{12}} \vee \cdots \vee G_m^{\pi_{1m}}\right)\right|,$$

*where const. depends only on $p$, $s$ and $G_1, \cdots, G_m$.*

*Proof.* Notice that

$$\mathbb{P}\left(G_1, \cdots, G_m \mid \pi_{12}^*, \cdots, \pi_{1m}^*\right) = \prod_{e \in \binom{[n]}{2}} \mathbb{P}\left(G_1(e), G_2(\pi_{12}^*(e)) \cdots, G_m(\pi_{1m}^*(e)) \mid \pi_{12}^*, \cdots, \pi_{1m}^*\right)$$

$$= \prod_{e \in \binom{[n]}{2}} \mathbb{P}\left(G_1(e), G_2'(e) \cdots, G_m'(e)\right) \tag{3}$$

where for a node pair $e = \{u, v\}$, the shorthand $\pi(e)$ denotes $\{\pi(u), \pi(v)\}$. The edge status of any node pair $e$ in the graph tuple $(G_1, G_2', \cdots, G_m')$ can be any of the $2^m$ bit strings of length $m$, but the corresponding probability in (3) depends only on the number of ones and zeros in the bit string. For $i \in [m]$, let $\alpha_i$ denote the number of node pairs $e$ whose corresponding tuple $(G_1(e), G_2'(e), \cdots, G_m'(e))$ has exactly $i$ 1's:

$$\alpha_i := \sum_{e \in \binom{[n]}{2}} \mathbb{1}\left\{(G_1(e), G_2'(e), \cdots, G_m'(e)) \text{ has exactly } i \text{ 1's}\right\}.$$

Two key observations are in order. First, it follows by definition that $\alpha_0 + \alpha_1 + \cdots + \alpha_m = \binom{n}{2}$. Second, by definition of $\alpha_i$, it follows that

$$\sum_{i=0}^{m} i\alpha_i = \sum_{e \in \binom{[n]}{2}} \sum_{j=1}^{m} G_j(e) = \sum_{e \in \binom{[n]}{2}} \sum_{j=1}^{m} G_j'(e) \tag{4}$$

is constant, independent of $\pi_{12}^*, \cdots, \pi_{1m}^*$. It follows then that

$$(3) = \left(1 - p + p(1-s)^m\right)^{\alpha_0} \times \prod_{i=1}^{m} \left(ps^i(1-s)^{m-i}\right)^{\alpha_i}$$

$$= \left(1 - p + p(1-s)^m\right)^{\alpha_0} \times p^{\sum_{i=1}^{m} \alpha_i} \times \prod_{i=1}^{m} \left(s^i(1-s)^{m-i}\right)^{\alpha_i}$$

$$= \left(1 - p + p(1-s)^m\right)^{\alpha_0} \times p^{\binom{n}{2} - \alpha_0} \times \left(\frac{s}{1-s}\right)^{\sum_{i=1}^{m} i\alpha_i} \times (1-s)^{m \sum_{i=1}^{m} \alpha_i}$$

$$= \left(\frac{1 - p + p(1-s)^m}{p(1-s)^m}\right)^{\alpha_0} \times (p(1-s)^m)^{\binom{n}{2}} \times \left(\frac{s}{1-s}\right)^{\sum_{i=1}^{m} i\alpha_i}$$

$$\propto \left(1 + \frac{1-p}{p(1-s)^m}\right)^{\alpha_0},$$

where the last step uses (4). Finally, since $\frac{1-p}{p(1-s)^m} > 0$, it follows that the log-likelihood satisfies

$$\log\left(\mathbb{P}\left(G_1, \cdots, G_m \mid \pi_{12}^*, \cdots, \pi_{1m}^*\right)\right) \propto \text{const.} + \alpha_0,$$

i.e. maximizing the likelihood corresponds to selecting $\pi_{12}, \cdots, \pi_{1m}$ to maximize $\alpha_0$ - the number of node pairs $e$ for which $G_1(e) = G_2(\pi_{12}(e)) = \cdots = G_m(\pi_{1m}(e)) = 0$. This is equivalent to minimizing the number of edges in the union graph $G_1 \vee G_2^{\pi_{12}} \vee \cdots \vee G_m^{\pi_{1m}}$, as desired. $\qquad \square$

**Remark 12.** *In the case of two graphs, minimizing the number of edges in the union graph $G_1 \vee_\pi G_2$ is equivalent to maximizing the number of edges in the intersection graph $G_1 \wedge_\pi G_2$. This is consistent with existing literature on two graphs [CK16, WXY22].*

## B   Concentration Inequalities for Binomial Random Variables

The following bounds for the binomial distribution are used frequently in the analysis.

**Lemma 13.** *Let $X \sim \mathsf{Bin}(n, p)$. Then,*

*1. For any $\delta > 0$,*

$$\mathbb{P}\left(X \geq (1+\delta)np\right) \leq \left(\frac{e^{\delta}}{(1+\delta)^{1+\delta}}\right)^{np} \leq \left(\frac{e}{1+\delta}\right)^{(1+\delta)np}. \tag{5}$$

*2. For any $\delta > 5$,*

$$\mathbb{P}\left(X \geq (1+\delta)np\right) \leq 2^{-(1+\delta)np}. \tag{6}$$

*3. For any $\delta \in (0, 1)$,*

$$\mathbb{P}\left(X \leq (1-\delta)np\right) \leq \left(\frac{e^{-\delta}}{(1-\delta)^{1-\delta}}\right)^{np}. \tag{7}$$

*Proof.* All proofs follow from the Chernoff bound and can be found, or easily derived, from Theorems 4.4 and 4.5 of [MU17]. □

## C   Proof of Lemma 8

We restate Lemma 8 for convenience.

**Lemma 8.** *Let $n$ and $k$ be positive integers and let $G \sim \mathsf{ER}(n, \alpha \log(n)/n)$ for some $\alpha > 0$. Let $v$ be a node of $G$ and let $\delta_G(v)$ denote the degree of $v$ in $G$. Then,*

$$\mathbb{P}\left(\{v \notin \mathsf{core}_k(G)\} \cap \{\delta_G(v) \geq k + 1/\alpha\}\right) = o\left(1/n\right). \tag{1}$$

Before presenting the proof, we present the intuition behind it. The events $\{v \notin \mathsf{core}_k(G)\}$ and $\{\delta_G(v) \geq k + 1/\alpha\}$ are highly negatively correlated. However, consider the subgraph $(G - v)$ of $G$ induced on the vertex set $V - \{v\}$, and note that the $k$-core of this subgraph does not depend on the degree of $v$. Furthermore, if $v \notin \mathsf{core}_k(G)$, then it must be that $v$ has fewer than $k$ neighbors in $\mathsf{core}_k(G - v)$. Intuitively, this event has low probability if $\mathsf{core}_k(G - v)$ is sufficiently large.

Notice that $(G - v) \sim \mathsf{ER}(n - 1, \alpha \log(n)/n)$, and so standard results about the size of the $k$-core of Erdős-Rényi graphs apply. However, we require the error probability that the $k$-core of $G - v$ is too small to be $o(1/n)$ - this is crucial since we will later use a union bound over all the nodes $v$. Unfortunately, standard results such as [Łuc91] can only be invoked directly to show that the corresponding probability is $o(1)$, which is insufficient for our purpose. Later in this section, we refine the analysis in [Łuc91] to obtain the desired convergence rate. The refinement culminates in the following.

**Lemma 14.** *Let $\alpha > 0$ and $G \sim \mathsf{ER}(n, \alpha \log(n)/n)$. Let $v$ be a node of $G$. The size of the $k$-core of $G - v$ satisfies*

$$\mathbb{P}\left(|\mathsf{core}_k(G - v)| < n - 3n^{1-\alpha}\right) = o(1/n).$$

The proof of Lemma 14 is deferred to Appendix C.1. It remains to study the error event that $v$ has too few neighbors in $\mathsf{core}_k(G - v)$. To count the number of neighbors of $v$ in $\mathsf{core}_k(G - v)$, we exploit the independence of $\mathsf{core}_k(G - v)$ and $v$ as follows: each neighbor of $v$ is considered a *success* if it belongs to $\mathsf{core}_k(G - v)$ and a *failure* otherwise. Counting the number of successes is equivalent to sampling *with* replacement $\delta_G(v)$ elements, each of which is independently a success with probability $|\mathsf{core}_k(G - v)|/(n - 1)$. The number of successes then follows precisely a hypergeometric distribution. This intuition is made rigorous in the proof below.

Let us recall some facts about the hypergeometric distribution because it plays an important role in the proof. Denote by $\mathsf{HypGeom}(N, K, n)$ a random variable that counts the number of successes in a sample of $n$ elements drawn *without replacement* from a population of $N$ individuals, of which

$K$ elements are considered successes. Note that if this sampling were done *with replacement*, then the number of successes would follow a $\mathsf{Bin}\,(n, K/N)$ distribution. A result of Hoeffding [Hoe94] establishes that the $\mathsf{HypGeom}(N, K, n)$ distribution is convex-order dominated by the $\mathsf{Bin}\,(n, K/N)$ distribution, i.e.

$$\mathbb{E}\left[f(\mathsf{HypGeom}(N, K, n))\right] \leq \mathbb{E}\left[f(\mathsf{Bin}(n, K/N))\right] \quad \text{for all convex functions } f.$$

In particular, the function $f(x) = e^{tx}$ is convex for any value of $t$, and so Chernoff bounds that hold for the binomial distribution also hold for the corresponding hypergeometric distribution. This yields the following proposition.

**Proposition 15.** *Let $X \sim \mathsf{HypGeom}(N, K, n)$. It follows for any $\delta > 0$ that*

$$\mathbb{P}\left(X > (1 + \delta) \times \frac{nK}{N}\right) \leq \left(\frac{e^\delta}{(1 + \delta)^{1+\delta}}\right)^{nK/N} \leq \left(\frac{e}{1 + \delta}\right)^{(1+\delta)nK/N}.$$

Our final remark about the hypergeometric distribution is a symmetry property. By interchanging the success and failure states, it follows that

$$\mathbb{P}\left(\mathsf{HypGeom}(N, K, n) = k\right) = \mathbb{P}\left(\mathsf{HypGeom}(N, N - K, n) = n - k\right).$$

The above intuition for the proof of Lemma 8 is formalized below.

*Proof of Lemma 8.* Let $V$ denote the vertex set of $G$, and let $G - v$ denote the induced subgraph of $G$ on the vertex set $V - \{v\}$. For any set $A \subseteq V$, let $N_v(A)$ denote the set of neighbors of $v$ in the set $A$, i.e.

$$N_v(A) := \{u \in A : \{u, v\} \in E(G)\}.$$

Since $\mathsf{core}_k(G - v) \subseteq \mathsf{core}_k(G)$, it is true that

$$\{v \notin \mathsf{core}_k(G)\} \subseteq \{|N_v(\mathsf{core}_k(G))| \leq k - 1\} \subseteq \{|N_v(\mathsf{core}_k(G - v))| \leq k - 1\}.$$

It follows that

$$\mathbb{P}\left(\{v \notin \mathsf{core}_k(G)\} \cap \{\delta_G(v) \geq k + 1/\alpha\}\right) \leq p_1 + p_2,$$

where

$$p_1 = \mathbb{P}\left(\{N_v(\mathsf{core}_k(G - v)) \leq k - 1\} \cap \{\delta_G(v) \geq k + 1/\alpha\} \cap \{|\mathsf{core}_k(G - v)| < n - 3n^{1-\alpha}\}\right)$$

$$p_2 = \mathbb{P}\left(\{N_v(\mathsf{core}_k(G - v)) \leq k - 1\} \cap \{\delta_G(v) \geq k + 1/\alpha\} \cap \{|\mathsf{core}_k(G - v)| \geq n - 3n^{1-\alpha}\}\right)$$

It suffices to show that both $p_1$ and $p_2$ are $o(1/n)$. The term $p_1$ deals with the probability that the $k$-core of $G - v$ is too small. In fact, by Lemma 14, it follows directly that

$$p_1 \leq \mathbb{P}\left(|\mathsf{core}_k(G - v)| < n - 3n^{1-\alpha}\right) = o(1/n),$$

Next, the probability $p_2$ is analyzed. Enumerate arbitrarily but independently the elements of sets $N_v(V)$ and $\mathsf{core}_k(G - v)$, so that

$$N_v(V) = \left\{v_1, \cdots, v_{\delta_G(v)}\right\}, \quad \mathsf{core}_k(G - v) = \left\{a_1, \cdots, a_{|\mathsf{core}_k(G-v)|}\right\}.$$

Given that $N_v(V)$ has more than $k + 1/\alpha$ nodes and $\mathsf{core}_k(G - v)$ has more than $n - 3n^{1-\alpha}$ nodes, it is true that

$$N_v(\mathsf{core}_k(G - v)) = N_v(V) \cap \mathsf{core}_k(G - v)$$

$$\supseteq \left\{v_1, \cdots, v_{\lceil k+1/\alpha \rceil}\right\} \cap \left\{a_1, \cdots, a_{\lceil n-3n^{1-\alpha} \rceil}\right\} =: \widetilde{N}_v(\widetilde{\mathsf{core}}_k(G - v)).$$

In words, $\widetilde{N}_v(\widetilde{\mathsf{core}}_k(G - v))$ counts among the first $\lceil k + 1/\alpha \rceil$ neighbors of $v$ those nodes that are also in the first $\lceil n - 3n^{1-\alpha} \rceil$ nodes of $\mathsf{core}_k(G - v)$. Therefore,

$$p_2 = \mathbb{P}\left(\{N_v(\mathsf{core}_k(G - v)) \leq k - 1\} \cap \{\delta_G(v) \geq k + 1/\alpha\} \cap \{|\mathsf{core}_k(G - v)| \geq n - 3n^{1-\alpha}\}\right)$$

$$\leq \mathbb{P}\left(\{N_v(\mathsf{core}_k(G - v)) \leq k - 1\} \mid \delta_G(v) \geq k + 1/\alpha, \ |\mathsf{core}_k(G - v)| \geq n - 3n^{1-\alpha}\right)$$

$$\leq \mathbb{P}\left(\{\widetilde{N}_v(\widetilde{\mathsf{core}}_k(G - v)) \leq k - 1\} \mid \delta_G(v) \geq k + 1/\alpha, \ |\mathsf{core}_k(G - v)| \geq n - 3n^{1-\alpha}\right) \quad (8)$$

Note that $\mathrm{core}_k(G - v)$ is entirely determined by the graph $G - v$, i.e. it is independent of the neighbors of $v$. Consequently, the two sets $\{v_1, \cdots, v_{\lceil k+1/\alpha \rceil}\}$ and $\{a_1, \cdots, a_{\lceil n-3n^{1-\alpha} \rceil}\}$ are selected independent of each other. Equivalently, given that $|\mathrm{core}_k(G - v)| \geq n - 3n^{1-\alpha}$ and $\delta_G(v) \geq k + 1/\alpha$, the size of the intersection set $\widetilde{N}_v(\widetilde{\mathrm{core}}_k(G - v))$ follows a hypergeometric distribution with parameters $(n - 1, \lceil n - 3n^{1-\alpha} \rceil, \lceil k + 1/\alpha \rceil)$. Therefore,

$$(8) = \mathbb{P}\left(\mathsf{HypGeom}(n - 1, \lceil n - 3n^{1-\alpha} \rceil, \lceil k + 1/\alpha \rceil) \leq k - 1\right)$$

$$\overset{(a)}{=} \mathbb{P}\left(\mathsf{HypGeom}(n - 1, n - 1 - \lceil n - 3n^{1-\alpha} \rceil, \lceil k + 1/\alpha \rceil) \geq \lceil k + 1/\alpha \rceil - (k - 1)\right)$$

$$= \mathbb{P}\left(\mathsf{HypGeom}(n - 1, \lfloor 3n^{1-\alpha} \rfloor - 1, \lceil k + 1/\alpha \rceil) \geq 1 + 1/\alpha\right) \tag{9}$$

where (a) uses the symmetry of the hypergeometric distribution. Using Proposition 15 and the fact that $n - 1 \geq n/2$ for any $n \geq 1$ yields

$$(9) \leq \left(e \cdot \frac{\lceil k + 1/\alpha \rceil}{1 + 1/\alpha} \cdot \frac{\lfloor 3n^{1-\alpha} \rfloor}{n - 1}\right)^{1+1/\alpha}$$

$$\leq \left(\frac{6e\lceil k + 1/\alpha \rceil}{1 + 1/\alpha}\right)^{1+1/\alpha} \times n^{-1-\alpha}$$

$$= o(1/n),$$

whenever $\alpha > 0$. $\qquad\square$

## C.1 Proof of Lemma 14

A key ingredient towards proving Lemma 14 is a useful result about the number of low-degree vertices in an Erdős-Rényi graph, presented next.

**Proposition 16.** *Let $\alpha > 0$ and $G \sim \mathsf{ER}\left(n - 1, \alpha \log(n)/n\right)$. Let $r$ be a positive integer and let $Z_r$ denote the set of vertices in $G$ with degree no more than $r$, i.e.*

$$Z_r = \{v \in V(G) : \delta_G(v) \leq r\}.$$

*For any $\delta$ such that $\delta > 1 - \alpha$, it is true that*

$$\mathbb{P}\left(|Z_r| \geq n^\delta\right) = o(1/n).$$

*Proof.* Notice that

$$\mathbb{P}\left(|Z_r| \geq n^\delta\right) = \mathbb{P}\left(\exists S' \subseteq V : \{|S'| \geq n^\delta\} \cap \left\{\max_{i \in S'} \delta_G(i) \leq r\right\}\right)$$

$$\leq \mathbb{P}\left(\exists\, S \subseteq V : \{|S| = n^\delta\} \cap \left\{\max_{i \in S} \delta_G(i) \leq r\right\}\right)$$

$$\leq \mathbb{P}\left(\exists\, S \subseteq V : \{|S| = n^\delta\} \cap \left\{\sum_{i \in S} \delta_G(i) \leq r|S|\right\}\right). \tag{10}$$

If $|S| = n^\delta$, then the sum of degrees of vertices in $S$ is the total number of edges with exactly end point in $S$, plus twice the number of edges with both end points in $S$. There are exactly $\binom{|S|}{2} + |S|(n - 1 - |S|) \leq n^{1+\delta}$ such vertex pairs, and each of them independently has an edge with probability $\alpha \log(n)/n$. Therefore, a union bound over all possible choices of $S$ yields

$$(10) \leq \binom{n - 1}{n^\delta} \cdot \mathbb{P}\left(\mathsf{Bin}\left(n^{1+\delta}, \alpha \log(n)/n\right) + \mathsf{Bin}\left(n^{2\delta}/2, \alpha \log(n)/n\right) \leq rn^\delta\right)$$

$$\leq \binom{n - 1}{n^\delta} \cdot \mathbb{P}\left(\mathsf{Bin}\left(n^{1+\delta}, \alpha \log(n)/n\right) \leq rn^\delta\right)$$

$$\overset{(a)}{\leq} \left(\frac{ne}{n^\delta}\right)^{n^\delta} \times \left(\frac{\exp\left(r/(\alpha \log n) - 1\right)}{(r/(\alpha \log n))^{r/(\alpha \log n)}}\right)^{n^\delta \alpha \log(n)}$$

$$= o(1/n),$$

whenever $\delta > 1 - \alpha$ as desired. Note that (a) uses the Binomial concentration inequality (7) and the fact that $\binom{n-1}{k} \leq \binom{n}{k} \leq \left(\frac{ne}{k}\right)^k$. $\qquad\square$

---

**Algorithm 3:** Łuczak expansion

---

**require :** Graph $G$, Set $U \subseteq V(G)$.

**1** $U_0 \leftarrow U$
**2 for** $i = 0, 1, 2, 3, \cdots$ **do**
**3**   **if** *there exists $u \in V \setminus U_i$ such that $u$ has 3 or more neighbors in $U_i$* **then**
**4**     $U_{i+1} \leftarrow U_i \cup \{u\}$
**5**   **else**
**6**     **return** $U_i$
**7**   **end**
**8 end**

---

Our objective is to show that the $k$-core of $G - v$ is sufficiently large with probability $1 - o(1/n)$. To that end, consider Algorithm 3 to identify a subset of the $k$-core, originally proposed by Łuczak [Łuc91].

Note that the **for** loop eventually terminates - the set $V \setminus U_i$ is empty, for example when $i = n$ for any input set $U$. The key is to realize that the **for** loop terminates much faster when the input $U = Z_{k+1}$, i.e the set of vertices of the input graph $G$ whose degree is $k + 1$ or less. Furthermore, the complement of the set output by the algorithm is contained in the $k$-core. Formally,

**Lemma 17.** *Let $U_f$ be the output of Algorithm 3 with input graph $G - v$ and set $U = Z_{k+1}$. Then,*

(a) $U_f^c \subseteq \mathsf{core}_k(G - v)$.

(b) *For any $\delta > 1 - \alpha$,*

$$\mathbb{P}\left(|U_f| > 3n^\delta\right) = o(1/n).$$

*Proof.* (a) The proof is by construction: Since $U_f$ is obtained by adding exactly $f$ nodes to $U_0$, it follows that $U_f^c \subseteq U_0^c = Z_{k+1}^c$, so each node in $U_f^c$ has degree $k + 2$ or more in $G - v$. Further, each node in $U_f^c$ has at most 2 neighbors in $U_f$, else the **for** loop would not have terminated. Thus, the subgraph of $G - v$ induced on the set $U_f^c$ has minimum degree at least $k$, and the result follows.

(b) If $|U_f| > 3n^\delta$, then either $|U_0| > 3n^\delta$ or there is some $M$ in $\{0, 1, \cdots, f\}$ for which $|U_M| = 3n^\delta$. Therefore,

$$\mathbb{P}\left(|U_f| > 3n^\delta\right) \leq \mathbb{P}\left(|U_0| > 3n^\delta\right) + \mathbb{P}\left(\exists\, M \in \{0, 1, \cdots, f\} \text{ s.t. } |U_M| = 3n^\delta\right)$$
$$= o(1/n) + \underbrace{\mathbb{P}\left(\exists\, M \in \{0, 1, \cdots, f\} \text{ s.t. } |U_M| = 3n^\delta\right)}_{(\star)},$$

by Proposition 16. Note that each iteration $i = 0, 1, \cdots, M - 1$ of the **for** loop adds exactly 1 vertex and at least 3 edges to the subgraph of $G - v$ induced on $U_M$. Therefore, the induced subgraph $G|_{U_M}$ has $3n^\delta$ vertices and at least $3\left(|U_M| - |U_0|\right)$ edges. Thus,

$$(\star) \leq \mathbb{P}\left(\exists \text{ subgraph } H = (W, F) \text{ of } G - v \text{ s.t. } |W| = 3n^\delta \text{ and } |F| \geq 3\left(3n^\delta - |U_0|\right)\right)$$
$$\leq \mathbb{P}\left(|U_0| > n^\delta\right) + \mathbb{P}\left(\exists \text{ subgraph } H = (W, F) \text{ of } G - v \text{ s.t. } |W| = 3n^\delta \text{ and } |F| \geq 6n^\delta\right)$$
$$\leq o(1/n) + \underbrace{\binom{n}{3n^\delta} \cdot \mathbb{P}\left(\mathsf{Bin}\left(\binom{3n^\delta}{2}, \frac{\alpha \log(n)}{n}\right) > 6n^\delta\right)}_{(\star\star)},$$

where the last step uses Proposition 16 and a union bound over all possible choices of $W$. Finally, using the relation $\binom{n}{k} \leq \left(\frac{ne}{k}\right)^k$ and the concentration inequality (5) from Lemma 13 yields

$$
\begin{aligned}
(\star\star) &\leq \left(\frac{n^{1-\delta}e}{3}\right)^{3n^\delta} \mathbb{P}\left(\mathsf{Bin}\left(\frac{9n^{2\delta}}{2}, \frac{\alpha \log n}{n}\right) > 6n^\delta\right) \\
&\leq (n^{1-\delta})^{3n^\delta} \times \left(\frac{3\alpha e \log n}{4n^{1-\delta}}\right)^{6n^\delta} \\
&= \left(\frac{3\alpha e \log n}{4n^{(1-\delta)/2}}\right)^{6n^\delta} \\
&= o(1/n),
\end{aligned}
$$

whenever $0 < \delta < 1$. The result follows. $\qquad\square$

Finally, notice that Lemma 17 directly implies Lemma 14.

## D  Proof of Theorem 9

**Theorem 9.** *Let $G_1, \cdots, G_m$ be correlated graphs from the subsampling model with parameters $C$ and $s$. Let $v \in V$ and let $U$ be a vertex cut of $\{1, \cdots, m\}$ such that $|U| \leq \lfloor m/2 \rfloor$. Then,*

$$
\mathbb{P}\left(\{c_v(U) = 0\} \cap \left\{\widetilde{c}_v(U) > \frac{m^2}{4}\left(k + \frac{1}{Cs^2}\right)\right\}\right) = o(1/n). \tag{2}
$$

*Proof.* For any vertex cut $U$,

$$
\begin{aligned}
\left\{\widetilde{c}_v(U) > \frac{m^2}{4}\left(k + \frac{1}{Cs^2}\right)\right\} &\overset{(a)}{\subseteq} \left\{\widetilde{c}_v(U) > |U|(m - |U|)\left(k + \frac{1}{Cs^2}\right)\right\} \\
&= \left\{\sum_{i \in U}\sum_{j \in U^c} \delta_{G'_i \wedge G'_j}(v) > |U|(m-|U|)\left(k + \frac{1}{Cs^2}\right)\right\} \\
&\subseteq \bigcup_{i \in U}\bigcup_{j \in U^c}\left\{\delta_{G'_i \wedge G'_j}(v) > k + \frac{1}{Cs^2}\right\},
\end{aligned}
$$

where (a) uses the fact that the maximum of a set of a numbers is greater than or equal to the average. On the other hand

$$
\{c_v(U) = 0\} = \bigcap_{i \in U}\bigcap_{j \in U^c}\left\{v \notin \mathsf{core}_k(G'_i \wedge G'_j)\right\}.
$$

Let $p_1$ denote the probability in the LHS of (2). It follows from the union bound that

$$
p_1 \leq \sum_{i \in U}\sum_{j \in U^c} \mathbb{P}\left(\left\{v \notin \mathsf{core}_k(G'_i \wedge G'_j)\right\} \cap \left\{\delta_{G'_i \wedge G'_j}(v) > k + \frac{1}{Cs^2}\right\}\right) = o(1/n),
$$

since for any choice of $i$ and $j$, the graph $G'_i \wedge G'_j \sim \mathsf{ER}\left(n, Cs^2 \log(n)/n\right)$. $\qquad\square$

## E  On Stochastic Dominance: Proof of Theorem 10

The objective of this section is to build up to a proof of Theorem 10. We start by making a simple observation about products of Binomial random variables.

**Lemma 18.** *Let $X_1, \cdots, X_m \sim \mathsf{Bern}(s)$ be i.i.d. random variables, and let $B = X_1 + \cdots + X_m$ denote their sum. For each $\ell$ in $\{1, 2, \cdots, \lfloor m/2 \rfloor\}$, define*

$$
T_\ell = (X_1 + \cdots + X_\ell)(X_{\ell+1} + \cdots + X_m).
$$

*For any $\ell_1, \ell_2 \in \{1, 2, \cdots, \lfloor m/2 \rfloor\}$ such that $\ell_1 < \ell_2$, and for any $t \in \mathbb{R}$ and any $b \in \{0, 1, \cdots, m\}$,*

$$
\mathbb{P}\left(T_{\ell_1} > t \mid B = b\right) \leq \mathbb{P}\left(T_{\ell_2} > t \mid B = b\right). \tag{11}
$$

505 *Proof of Lemma 18.* Consider overlapping but exhaustive cases:

506 *Case 1: $t < 0$.* Since $T_\ell \geq 0$ almost surely for all $\ell$, the inequality (11) holds.

507 *Case 2: $t \geq b - 1$.* Note that conditioned on $B = b$, it follows that $T_1 \in \{0, b-1\}$. Therefore, the
508 left hand side of (11) equals zero, and the inequality holds.

509 *Case 3: $b = 0$ or $b = 1$.* In this case, $T_\ell$ is identically zero for all $\ell$, so (11) holds.

510 *Case 4: $b \geq 2$ and $0 \leq t < b - 1$.* For any $\ell \in \{1, 2, \cdots, \lfloor m/2 \rfloor\}$,

$$
\begin{aligned}
\mathbb{P}\left(T_\ell > t \mid B = b\right) &= \frac{\mathbb{P}\left(\{(X_1 + \cdots + X_\ell)(X_{\ell+1} + \cdots + X_m) > t\} \cap \{X_1 + \cdots + X_m = b\}\right)}{\mathbb{P}\left(X_1 + \cdots + X_m = b\right)} \\
&= \frac{\sum_{i:i(b-i)>t} \mathbb{P}\left(\{X_1 + \cdots + X_\ell = i\} \cap \{X_{\ell+1} + \cdots + X_m = b - i\}\right)}{\mathbb{P}\left(X_1 + \cdots + X_m = b\right)} \\
&\overset{(a)}{=} \frac{\sum_{i=1}^{b-1} \mathbb{P}\left(X_1 + \cdots + X_\ell = i\right) \mathbb{P}\left(X_{\ell+1} + \cdots + X_m = b - i\right)}{\mathbb{P}\left(X_1 + \cdots + X_m = b\right)} \\
&\overset{(b)}{=} \frac{\sum_{i=1}^{b-1} \binom{\ell}{i}\binom{m-\ell}{b-i}}{\binom{m}{b}} \\
&= \frac{\sum_{i=0}^{b} \binom{\ell}{i}\binom{m-\ell}{b-i} - \binom{m-\ell}{b} - \binom{\ell}{b}}{\binom{m}{b}} \\
&= \frac{\binom{m}{b} - \binom{m-\ell}{b} - \binom{\ell}{b}}{\binom{m}{b}},
\end{aligned}
\tag{12}
$$

where (a) used the fact that for any $t$ such that $0 \leq t < b - 1$, it is true that

$$
\{i : i(b - i) > t\} = \{1, 2, \cdots, b - 1\}.
$$

511 Here, the notation for binomial coefficients in (b) involves setting $\binom{n}{k} = 0$ whenever $k < 0$ or $k > n$.
512 Let $f_{m,b}(\ell)$ denote the numerator of (12), i.e.

$$
f_{m,b}(\ell) := \binom{m}{b} - \binom{m-\ell}{b} - \binom{\ell}{b}.
$$

513 It suffices to show that $f_{m,b}(\ell) - f_{m,b}(\ell - 1) \geq 0$ for all $\ell \in \{2, \cdots, \lfloor m/2 \rfloor\}$. Indeed,

$$
\begin{aligned}
f_{m,b}(\ell) - f_{m,b}(\ell - 1) &= \binom{m-\ell+1}{b} - \binom{m-\ell}{b} - \left(\binom{\ell}{b} - \binom{\ell-1}{b}\right) \\
&\overset{(c)}{=} \binom{m-\ell}{b-1} - \binom{\ell-1}{b-1} \geq 0,
\end{aligned}
$$

514 whenever $m - \ell \geq \ell - 1$, i.e. $\ell \leq \lfloor m/2 \rfloor$. Here, (c) uses the identity $\binom{n}{k} = \binom{n-1}{k-1} + \binom{n-1}{k}$, and the
515 fact that $\binom{n_1}{k} \geq \binom{n_2}{k}$ whenever $n_1 \geq n_2$. This concludes the proof. $\qquad\square$

516 **Corollary 19.** *Let $F$ be a collection of edges in the parent graph $G$. For any edge $e_r \in F$, let $X_i^r$*
517 *denote the indicator random variable $G_i'(e_r) \sim \text{Bern}(ps)$. For each $\ell$ in $\{1, \cdots, \lfloor m/2 \rfloor\}$, define*

$$
T_\ell^r = (X_1^r + \cdots + X_\ell^r)(X_{\ell+1}^r + \cdots + X_m^r).
$$

518 *Then, for any $\ell_1, \ell_2 \in \{1, \cdots, \lfloor m/2 \rfloor\}$ such that $\ell_1 < \ell_2$, the following stochastic ordering holds*

$$
\sum_{r=1}^{|F|} T_{\ell_1}^r \preceq \sum_{r=1}^{|F|} T_{\ell_2}^r.
$$

519 *Proof.* It suffices to show that $T_{\ell_1}^r \preceq T_{\ell_2}^r$ for each $r$, since the edges are independent. Indeed, we
520 have for any $t$ that

$$
\mathbb{P}\left(T_{\ell_1}^r > t\right) = \sum_{b=0}^{m} \mathbb{P}(B = b) \mathbb{P}\left(T_{\ell_1}^r > t \mid B = b\right) \leq \sum_{b=0}^{m} \mathbb{P}(B = b) \mathbb{P}\left(T_{\ell_2}^r > t \mid B = b\right) = \mathbb{P}\left(T_{\ell_2}^r > t\right),
$$

521 which concludes the proof. $\qquad\square$

522   With this, we are ready to prove Theorem 10. The theorem is restated for convenience.

523   **Theorem 10.** *Let $G_1, \cdots, G_m$ be correlated graphs from the subsampling model. Let $v \in V$ and*
524   *let $U_\ell$ denote the set $\{1, \cdots, \ell\}$ for $\ell$ in $\{1, \cdots, \lfloor m/2 \rfloor\}$. For any vertex cut $U$ of $\{1, \cdots, m\}$, let*
525   *$\widetilde{c}_v(U)$ denote its cost in the graph $\widetilde{\mathcal{H}}(v)$. The following stochastic ordering holds:*

$$\widetilde{c}_v(U_1) \preceq \widetilde{c}_v(U_2) \preceq \cdots \preceq \widetilde{c}_v(U_{\lfloor m/2 \rfloor}).$$

526   *Proof.* Let $\ell_1, \ell_2 \in \{1, \cdots, \lfloor m/2 \rfloor\}$ such that $\ell_1 < \ell_2$. Let $t \in \mathbb{R}$. Consider the parent graph $G$ and
527   label the set of incident edges on $v$ as $\{e_1, \cdots, e_{\delta_G(v)}\}$. Denote by $X_i^r$ the indicator random variable
528   $G_i'(e_r) \sim \text{Bern}(ps)$. It follows that

$$\mathbb{P}\left(\widetilde{c}_v(U_{\ell_2}) > t\right) = \mathbb{P}\left(\sum_{i=1}^{\ell_2} \sum_{j=\ell_2+1}^{m} \delta_{G_i' \wedge G_j'}(v) \geq t\right)$$

$$= \mathbb{P}\left(\sum_{i=1}^{\ell_2} \sum_{j=\ell_2+1}^{m} \sum_{r=1}^{\delta_G(v)} X_i^r X_j^r > t\right)$$

$$= \sum_{d=0}^{n} \mathbb{P}\left(\delta_G(v) = d\right) \mathbb{P}\left(\sum_{r=1}^{d} \left((X_1^r + \cdots + X_{\ell_2}^r)(X_{\ell_2+1}^r + \cdots + X_m^r)\right) > t\right)$$

$$\overset{(a)}{\geq} \sum_{d=0}^{n} \mathbb{P}\left(\delta_G(v) = d\right) \mathbb{P}\left(\sum_{r=1}^{d} \left((X_1^r + \cdots + X_{\ell_1}^r)(X_{\ell_1+1}^r + \cdots + X_m^r)\right) > t\right)$$

$$= \mathbb{P}\left(\sum_{i=1}^{\ell_1} \sum_{j=\ell_1+1}^{m} \sum_{r=1}^{\delta_G(v)} X_i^r X_j^r > t\right)$$

$$= \mathbb{P}\left(\sum_{i=1}^{\ell_1} \sum_{j=\ell_1+1}^{m} \delta_{G_i' \wedge G_j'}(v) \geq t\right)$$

$$= \mathbb{P}\left(\widetilde{c}_v(U_{\ell_1}) > t\right),$$

529   as desired. Here, (a) uses Corollary 19. $\qquad \square$

## F   On Low Degree Nodes: Proof of Theorem 11

531   **Theorem 11.** *Let $G_1, \cdots, G_m$ be obtained from the subsampling model with parameters $C$ and $s$.*
532   *Let $r = \frac{m^2}{4}\left(k + \frac{1}{Cs^2}\right)$. Let $v \in [n]$ and suppose that $Cs(1 - (1-s)^{m-1}) > 1$. Then,*

$$\mathbb{P}\left(\widetilde{c}_v(U_1) \leq r\right) \leq \mathbb{P}\left(\left\{\delta_{G_1 \wedge G_2'}(v) \leq r\right\} \cap \cdots \cap \left\{\delta_{G_1 \wedge G_m'}(v) \leq r\right\}\right) = o\left(1/n\right).$$

533   *Proof.* Consider fixed integers $r_1, \cdots, r_m$ such that $0 \leq r_2, \cdots, r_m \leq r$. Since $r$ is constant, by a
534   union bound argument it suffices to show

$$(\star) =: \mathbb{P}\left(\left\{\delta_{G_1 \wedge G_2'}(v) = r_2\right\} \cap \cdots \cap \left\{\delta_{G_1 \wedge G_m'}(v) = r_m\right\}\right) = o\left(1/n\right).$$

535   Proceed by conditioning on the degree of $v$ in $G_1$, which follows a $\text{Bin}(n, ps)$ distribution. Since
536   the degrees of $v$ in the intersection graphs $\{G_1 \wedge G_i' : i = 2, \cdots, m\}$ are conditionally independent
537   given the degree of $v$ in $G_1$, we have

$$(\star) = \mathbb{E}_D\left[\mathbb{P}\left(\bigcap_{i=2}^{m} \left\{\delta_{G_1 \wedge G_i'}(v) = r_i\right\} \Big| \delta_{G_1}(v) = D\right)\right]$$

$$= \mathbb{E}_D\left[\prod_{i=2}^{m} \mathbb{P}\left(\left\{\delta_{G_1 \wedge G_i'}(v) = r_i\right\} \Big| \delta_{G_1}(v) = D\right)\right]$$

$$= \mathbb{E}_D\left[\prod_{i=2}^{m} \binom{D}{r_i} s^{r_i}(1-s)^{D-r_i}\right]. \tag{13}$$

538    Using the fact that $\binom{D}{r_i} \leq \left(\frac{De}{r_i}\right)^{r_i}$, it follows that

$$(13) \leq \left(\frac{se}{1-s}\right)^{\sum_{i=2}^m r_i} \cdot \prod_{i=2}^m r_i^{-r_i} \times \mathbb{E}_D\left[D^{\sum_{i=2}^m r_i} \times (1-s)^{(m-1)D}\right]$$

$$\leq \text{const.} \times \mathbb{E}_D\left[D^{\sum_{i=2}^m r_i} \times (1-s)^{(m-1)D}\right]. \tag{14}$$

539    Expanding out the expectation yields

$$(14) = \text{const.} \times \sum_{d=0}^n L_d, \text{ where } L_d := d^{\sum_{i=2}^m r_i}(1-s)^{(m-1)d} \times \mathbb{P}\left(\mathsf{Bin}(n,ps) = d\right).$$

540    Proceed by splitting the summation at $(\log n)^2$. The first part can be bounded as

$$\sum_{d=0}^{(\log n)^2} L_d \leq (\log n)^{2\sum_{i=2}^m r_i} \sum_{d=0}^{(\log n)^2} (1-s)^{(m-1)d} \cdot \mathbb{P}\left(\mathsf{Bin}(n,ps) = d\right)$$

$$\leq (\log n)^{2\sum_{i=2}^m r_i} \cdot \sum_{d=0}^n (1-s)^{(m-1)d} \cdot \mathbb{P}\left(\mathsf{Bin}(n,ps) = d\right)$$

$$= (\log n)^{2\sum_{i=2}^m r_i} \cdot \mathbb{E}_D\left[(1-s)^{(m-1)D}\right]$$

$$\overset{(a)}{=} (\log n)^{2\sum_{i=2}^m r_i} \cdot \left(1 - \frac{Cs\left(1-(1-s)^{m-1}\right)\log n}{n}\right)^n$$

$$= o(1/n),$$

541    whenever $Cs(1-(1-s))^{m-1} > 1$. Here, (a) is obtained by evaluating the probability generating
542    function of the $\mathsf{Bin}(n,ps)$ random variable at $(1-s)^{m-1}$ and setting $p = C\log(n)/n$.

543    The other part of the sum can now be bounded as follows.

$$\sum_{d=(\log n)^2}^n L_d \leq \left[\max_{d:\, (\log n)^2 \leq d \leq n} d^{\sum_{i=2}^m r_i}(1-s)^{md}\right] \cdot \mathbb{P}\left(\mathsf{Bin}(n,ps) \geq (\log n)^2\right)$$

$$\overset{(b)}{\leq} \left[(\log n)^{2\sum_{i=2}^m r_i}(1-s)^{m(\log n)^2}\right] \times 2^{-(\log n)^2}$$

$$= (\log n)^{2\sum_{i=2}^m r_i}\left(\frac{(1-s)^m}{2}\right)^{(\log n)^2}$$

$$= o\left(1/n\right).$$

544    whenever $C > 0$. Here, (b) is true because the function $d \mapsto d^{\sum r_i}(a-s)^{md}$ is decreasing on the
545    interval $[(\log n)^2, n]$ for all sufficiently large $n$. Finally, the concentration inequality for the Binomial
546    distribution holds by (6) in Lemma 13. The inequality applies since $p = C\log(n)/n$ and since
547    $(\log n)^2 > 6Cs\log(n)$ for all $n$ sufficiently large. This concludes the proof. $\qquad\square$

