# OpenReview forum: "Exact Random Graph Matching with Multiple Graphs"
_NeurIPS.cc/2024/Conference — Submitted to NeurIPS 2024_

### Official Review · Reviewer_tQgp · 2024-06-20

**Soundness:** 4
**Presentation:** 3
**Contribution:** 4
**Rating:** 7
**Confidence:** 5

**Summary:**

This paper considers the graph matching problem, where the goal is to produce a mapping between vertices of multiple graphs which maximizes similarities among them. The authors study graph matching from a theoretical perspective, in which one observes multiple (appropriately correlated) Erdös-Rényi (ER) graphs that have ground-truth latent mappings between them. The authors' goal is to characterize the information-theoretic threshold for exactly recovering the latent mappings between all of the observed ER graphs. Prior work has settled the information-theoretic thresholds for 2 correlated ER graphs, and this paper settles it for more than 2 ER graphs.

To determine the information-theoretic threshold for exact graph matching, the authors establish matching achievability and converse results. The converse is based on a simple reduction to a graph matching problem with two ER graphs, combined with known results on impossibility results for exactly matching two ER graphs. For the achievability results, two algorithms are discussed. The first is the MLE, which is optimal for exact graph matching. The authors show that it has a clean, easy-to-understand form: the MLE outputs vertex mappings which maximize the number of edges in the corresponding union graph. However, the authors do not directly analyze this algorithm due to technical complexities. Instead, they propose an algorithm which involves two phases. (1) For each pair of graphs, a partial, fully-correct mapping is computed via the $k$-core estimator, and (2) unmatched vertices are matched through a "transitive closure" procedure. This algorithm provably outputs the full, correct set of vertex mappings in the parameter region that complements the converse.

Finally, a few numerical experiments are presented, showing that the transitive closure procedure can be combined with known computationally efficient algorithms for pairwise graph matching to derive algorithms for matching multiple graphs in a principled manner.

**Strengths:**

Almost all the existing theoretical work on graph matching concerns two graphs, except for some trivial results (to the best of my knowledge). The extension of the theoretical framework to multiple graphs is a natural and important follow-up, and may inspire several future works as well.

While the algorithms and analysis are largely adapted from prior work (e.g., the $k$-core estimator), a key novelty is the transitive closure step, which provides a principled (and optimal!) bridge between pairwise graph matching and $m$-ary graph matching. As the authors highlight, this step can be used to extend practical algorithms for pairwise matching to the $m$-ary case in a black-box manner. I imagine that this technique could be useful in practice.

Additionally, the paper is well-written.

**Weaknesses:**

To me, the main weakness of the paper is in the discussion of transitive closure's implications. The authors make a striking observation that one can use their transitive closure technique (at least heuristically) to generalize pairwise graph matching to $m$-ary graph matching. However, several details are lacking in the simulations section. For instance, what are the graph parameters ($n$ and $p$)? What is the error rate before and after the transitive closure boosting? How do the results shown compare to the accuracy of Algorithm 2? (Even though the $k$-core matching is not efficient to compute, the result of the matching procedure is a function of the ground-truth permutations, so I believe the algorithm's accuracy can be simulated efficiently).

There are a couple other minor weaknesses. One is that Algorithm 2 is computationally inefficient. However, making such an algorithm efficient is likely a challenging research question itself, and is appropriate for future work. Another weakness is that there is no nice figure to visualize Algorithm 2. I feel that the reader's understanding could be greatly improved if one could create a representative figure for the transitive closure boosting.

**Questions:**

Here I've listed a number of minor questions / comments in addition to the weaknesses above.
- (p. 1, line 21) I'm not sure if it's clear at this point what "latent correspondence" means. Perhaps motivate this notion through social networks first, where the latent correspondence is derived from the same users across the two networks.
- (p. 1, line 30) I'd suggest adding a sentence or two at the end of the introduction summarizing your contributions. But this is more of a stylistic comment.
- (p. 2, line 39) I'm not sure if it's clear at this point what "matching all vertices correctly" means. I think you mean to say something along the lines of *exactly* learning the *entire* latent correspondence. (I'm just worried that using *matching* and *correspondence* without clarification of equivalence could confuse readers outside of this domain.)
- (p. 2, line 50) I would mention that the literature on establishing information-theoretic limits largely (including the $k$-core estimator) largely uses computationally inefficient algorithms.
- (p. 3, line 93) I would emphasize that permuting node labels is equivalent to removing node-level information. Hence any algorithm must use topological features, rather than node-level features, to do matching in this setting.
- (p. 3, line 106) Emphasize that matchings need not be complete, for clarity.
- (p. 4, line 116) I think you mean that $Cs (1 - (1 - s)^{m - 1}) \ge 1$ is a necessary condition. Similarly, in the next line you should say something like "this condition is also sufficient (except for the equality case) to exactly match $m$ graphs with probability going to 1."
- (p. 4, line 139) Can you say something about *why* the analysis of the MLE is cumbersome?
- (p. 5, Algorithm 1) Is the algorithm environment here really necessary? Maybe for brevity (and for minimal confusion) you can simply write, in the main text,
$$
(\widehat{\pi}_{12}^{ML}, \ldots, \widehat{\pi}_{1m}^{ML}) \in argmax |E(G_1 \lor G_2^{\pi_{12}} \lor \ldots \lor G_m^{\pi_{1m}})|
$$
- (p. 5, line 146) The sentence "If a node $v$ is unmatched..." could really use an illustrative figure, as this idea is really the essence of transitive closure boosting. Also, I would add for clarity that the sequence of graphs must consist of unique elements, and need not use all the graphs.
- (p. 7, Definition 7) In light of Lemma 8, it seems that $\mathcal{H}(v)$ can be derived from $\widetilde{\mathcal{H}}(v)$ by just thresholding the edge weights. Is there a strong reason why the analysis is done on the (seemingly more complex) weighted graph?
- (p. 7, Theorem 9) What's the intuition behind the threshold $\frac{m^2}{4} \left( k + \frac{1}{Cs^2} \right)$?
- (p. 9) Perhaps put the simulations and discussion of transitive closure into its own section.
- (p. 9, bullet point on Beyond ER graphs) The theoretical literature on the $k$-core estimator allows for quite general classes of random graphs. What specific aspects of your analysis would be challenging to extend beyond ER?
- (general comment) For two graphs, the information-theoretic threshold has a nice qualitative interpretation: it is the connectivity threshold of the common (intersection) graph. Is there a similar qualitative interpretation for the $m$-ary graph matching threshold?
- (general comment) It would be useful to note that when $s$ is small, the information-theoretic threshold is reduced by a multiplicative factor of $m -1$, compared to the $m = 2$ case. This follows since $1 - (1 - s)^{m - 1} \approx (m - 1) s$ for small $s$.

**Limitations:**

Limitations have been largely discussed. The authors could expand upon implications of graph matching to protecting / breaking privacy in anonymized social networks.

---

> ### Author Rebuttal · Authors · 2024-08-07
>
> Thank you for your review. Our original submission uses $(n,p) = (10^3,0.1)$ in the simulation, and we have now included details such as error rates before and after boosting through transitive closure in the PDF file accompanying our global response. These results are for the simulated output of the $k$-core estimator, which can be efficiently obtained as you pointed out. Please see the global response for some remarks on this.
>
> Thank you also for the many suggestions on improving the presentation of the paper. We will make sure to incorporate these if the paper is accepted. Below, we would like to address your questions.
>
> **On analyzing the MLE:** For any set of permutations $(\pi_{12},\cdots,\pi_{1m})$, the random variable $X(\pi_{12},\cdots,\pi_{1m})$ of interest is the number of edges in the corresponding union graph $G_1 \vee G_2^{\pi_{12}} \vee \cdots \vee G_m^{\pi_{1m}}$. When $m=2$, it is possible to obtain MGF bounds for $X(\pi)$ based on the orbit decomposition of $\pi$ and the number of correctly matched nodes in $\pi$, but we are unable to obtain an analogous result for general $m$. The difficulty is in getting a handle on how different orbits in the decompositions of $\pi_{12},\cdots,\pi_{1m}$ interact with each other to determine the distribution of $X$.
>
> **On using the weighted graph $\widetilde{\mathcal{H}}(v)$**: We chose the weighted graph representation because our analysis uses the fact that for any vertex cut in $\widetilde{\mathcal{H}}(v)$: if the *sum* of the weights of all edges crossing the cut is larger than a threshold $\tau$, then the graph $\mathcal{H}(v)$ must have at least one crossing edge for the same cut. This allows us to threshold the sum of the weights over all edges, rather than the individual weights of each edge, and the former statistic is easier to compare across different cuts.
>
> **Intuition behind the threshold $\frac{m^2}{4}\left( k + \frac{1}{Cs^2}\right)$:** Our choice of this threshold is an artifact of the analysis. If the total crossing cost in $\widetilde{\mathcal{H}}(v)$ for a $1$-cut is larger than $\frac{m^2}{4}\left(k+\frac{1}{Cs^2}\right)$, then the total crossing cost for any other cut (which has stochastically larger crossing cost) is also larger than $\frac{m^2}{4}\left(k+\frac{1}{Cs^2}\right)$. The extreme case in an $m/2$ cut, for which there are $m^2/4$ possible edges that cross the cut: our threshold choice ensures that even in this extreme case there is a crossing edge $(i,j)$ in $\widetilde{\mathcal{H}}(v)$ whose cost is larger than $\left(k+\frac{1}{Cs^2}\right)$, so the corresponding crossing edge must exist in $\mathcal{H}(v)$. This argument rules out all possible cuts and allows us to conclude that $\mathcal{H}(v)$ must be connected under our necessary condition. Note that any threshold larger than our choice would also work for this argument.
>
> **On extending the framework beyond ER graphs**: Indeed, the $k$-core estimator works well for pairwise graph matching in general settings such as inhomogeneous random graphs [RS23]. The bottleneck is to prove an analogous result to Lemma 8, i.e. sufficient condition for whether a node is included in the $k$-core of a graph based on its degree. Our proof of Lemma 8 crucially exploits the fact that: given the size of the $k$-core of an ER graph $G$, the set of nodes that form the $k$-core are all equally likely. This symmetry is lost in non-ER models. Even so, the analysis may be tractable for inhomogeneous graphs - we leave this for future work.
>
> **On interpretation via connectivity threshold:** By the proof of the impossibility result (Theorem 2), the threshold $Cs(1-(1-s)^{m-1})$ can be viewed as the connectivity threshold of the intersection graph between $G_m$ and the union of the other $m-1$ graphs.

---

> > ### Comment · Reviewer_tQgp · 2024-08-12
> >
> > Thanks for the detailed response and further discussion of transitive closure! I will maintain my positive score.

---

### Official Review · Reviewer_T9Pj · 2024-07-05

**Soundness:** 4
**Presentation:** 4
**Contribution:** 4
**Rating:** 8
**Confidence:** 4

**Summary:**

This paper studies the information theoretic limits for matching
multiple correlated random graphs. Based on a correlated Erdos-Renyi
random graph model, the authors provide both lower bound and achievable
bound for the condition to correctly match all nodes with high
probability. These bounds match each other. A highly interesting insight
is that, even when exactly matching two graphs is not possible, the
proposed algorithm can leverage more than two graphs to produce exact
matching among all the graphs. The achievable algorithm exploits the
transitivity among partial matchings through $k$-cores, which is also
quite interesting.

**Strengths:**

1. The novelty of the paper is high in dealing with graph matching among
multiple correlated graphs.

2. The necessary and sufficient conditions for exact matching meet each
other.

3. The proposed algorithm can exploit transitivity to match all graphs,
even when any two graphs alone cannot be exactly matched. This is a very
insightful result.

**Weaknesses:**

1. The proposed algorithms do incur high complexity.

**Questions:**

1. In Algorithm 2, first the $k$-core for each pair of graphs is found.
However, there is no discussion of how to choose the parameter $k$.

2. Does Proposition 5 depend on the sub-sampling probability $s$?

**Limitations:**

Limitations are discussed in Section 5.

---

> ### Author Rebuttal · Authors · 2024-08-07
>
> Thank you for your review. We address the two questions below:
>
> 1. The theoretical guarantees for the $k$-core estimator hold for any constant $k \geq 13$ (this is an artifact of the analysis). In the PDF file (global response) with simulation results, we use $k \in \lbrace 13,14 \rbrace$ because smaller values of $k$ correspond to larger $k$-cores of the graph.
>
> 2. The statement of Proposition 5 is true for any constant $s \in (0,1]$.

---

> > ### Comment · Reviewer_T9Pj · 2024-08-09
> >
> > Thank you for the response! I wish to maintain my current score.

---

### Official Review · Reviewer_Vr8o · 2024-07-14

**Soundness:** 4
**Presentation:** 4
**Contribution:** 4
**Rating:** 6
**Confidence:** 4

**Summary:**

This theoretical paper gives tight conditions for exact graph matching with multiple correlated random graphs. This problem has been extensively studied recently for the case of 2 graphs, and it is shown here that with more than 2 graphs, there is a regime where pairwise alignment is not possible, but with the information provided by all graphs, it is possible to align all of them. This is a nice theoretical result.

**Strengths:**

This paper studies a natural extension of a well-studied problem from 2 graphs to more graphs and shows a surprising effect: making partial pairwise matching is sufficient to get the exact recovery. The proof outlines give the main insights into the technical proof.

**Weaknesses:**

The resulting algorithm is not practical as it does not run in polynomial time (as it is mentioned by the authors).

**Questions:**

This paper is mostly theoretical and would probably benefit from being published in a more math-oriented journal. The impact on the Neurips community and the feedback here will probably be limited.

**Limitations:**

The authors are very clear with the limitations of their work in section 5.

---

> ### Author Rebuttal · Authors · 2024-08-07
>
> Thank you for your review. Please see our response to reviewer Yg8a for a note on contextualizing our work with respect to NeurIPS, and the PDF in our global response for experimental evaluation of our algorithm in ER and non-ER models.

---

> > ### Comment · Reviewer_Vr8o · 2024-08-09
> >
> > Thank you for your rebuttal. I still think your paper is a very solid contribution and would like to see it published in a more theoretical venue (which is not incompatible with a presentation at NeurIPS).

---

### Official Review · Reviewer_Yg8a · 2024-07-23

**Soundness:** 2
**Presentation:** 2
**Contribution:** 2
**Rating:** 4
**Confidence:** 3

**Summary:**

The paper aims to find out alignments between G_1 and G_2,....G_m, under the assumptions that they all are essentially sampled from ER graph distribution. The paper presents one impossibility result (or necessary condition to estimate such alignment)  and two sufficiency results to solve the underlying problem.

**Strengths:**

The paper tackles an interesting problem and it is written clearly.

**Weaknesses:**

(1) I am not too confident that the paper is appropriate for neurips audience. I think the paper suits better to a conference like ISIT or such.  The paper has barely any learning component and the practical utility is not very clear.
Also, the primary area assigned by the authors "Probabilistic methods (for example: variational inference, Gaussian processes)" is probably not correct.

(2) The paper only tackles a very simple graph model (ER graph model). While I understand that theoretical analysis for complex graph model is difficult, I would recommend the authors should discuss that in comprehensive manner. To elaborate concretely,
suppose,  G_1, G' _2...,G' _m are *not* generated from an ER model. But G_2,...G_m are generated using an ER like model with constant edge deletion probability $s$. In such case, can one characterize the necessary and sufficient condition.

Note that, the area is not too new in the literature. There has been work already in this line of research [CK17,WXY22 in the paper]. Although I will not say this work is an extension but the theoretical contribution given the existing works is not very interesting (m=2 to an arbitrary m for example).

(3) There is no experimental analysis. I would have increased my rating if the authors have done a thorough study on implications (including limitations) of their work on graphs from other models. For example, if we apply the same algorithm in other graph models, how would it perform. Since the line of work is not new, I would not say the theoretical results have strong enough impact to ignore the poor experiments.

**Questions:**

See above.

**Limitations:**

Restrictive graph model; poor experiments and incremental contribution.

---

> ### Author Rebuttal · Authors · 2024-08-07
>
> Thank you for your review.
>
> **On relevance to NeurIPS:** NeurIPS and other learning conferences have been the venue of choice for other works in graph matching that establish information-theoretic recovery limits in various settings, such as [1]-[4] below. Since graph matching is an important data processing step for various downstream machine learning tasks (for example in vision and NLP), and since our transitive closure step provides an efficient black-box approach to extend $2$-ary matchings to $m$-ary matchings, we feel that NeurIPS is a relevant venue for our work.
>
> **On theory and experimentation:** Our principal objective is a theoretical analysis for ER graphs to illustrate a key insight: pairwise matching followed by boosting is an optimal bridge between $2$-ary and $m$-ary graph matching. We leave the analysis of more general models such as stochastic block model and inhomogeneous random graphs to future work (please see our response to Reviewer tQgp for a comment on bottlenecks to this extension). Even so, we agree that experimental verification of this insight would bolster our work. In the attached PDF (global response), simulation results for ER graphs with high and low correlation, as well as a non-ER model (stochastic block model with 5 communities) is presented. We feel that these results are promising for the use of transitive closure as a subroutine to combine information from multiple graphs in practice. If the paper is accepted, we will use the extra page allowed in the camera-ready version to contextualize these simulation results.
>
> **References**
>
> [1] Racz, M., & Sridhar, A. (2021). Correlated stochastic block models: Exact graph matching with applications to recovering communities. In *Neural Information Processing Systems (Spotlight Paper)*, 34, 22259-22273.
>
> [2] Wang, Z., Wang, W., & Wang, L. (2024). Efficient Algorithms for Attributed Graph Alignment with Vanishing Edge Correlation. In  *Conference on Learning Theory* (pp. 4889-4890). PMLR.
>
> [3]  Ameen, T. & Hajek, B. (2024). Robust Graph Matching when Nodes are Corrupt. In *International Conference on Machine Learning*, (235: pp 1276-1305). PMLR.
>
> [4] Gaudio, J., Racz, M. Z., & Sridhar, A. (2022). Exact community recovery in correlated stochastic block models. In *Conference on Learning Theory* (pp. 2183-2241). PMLR.

---

> > ### Comment · Reviewer_Yg8a · 2024-08-14
> > **Response**
> >
> > I have read the rebuttal; but I still believe it may not attract the audience of Neurips. The results are OK. But, they are still on synthetic dataset. I am not convinced it's real world application yet. Though I have increased the score by +1, I am not too positive.

---

### Author Rebuttal · Authors · 2024-08-07

We would like to address experimental evaluation of our proposed algorithm in the global response, since this was raised by multiple reviewers.

As Reviewer tQgp pointed out: the $k$-core estimator is not efficient, but its output can be efficiently simulated by computing the $k$-core of the true intersection graph. Theoretical guarantees from [CKMP19, RS23, GRS22] establish that the output of the $k$-core estimator coincides with the $k$-core of the true intersection graph with probability $1-o(1)$ as $n\to\infty$, when $k \geq 13$ and and $p = \Theta(\log(n)/n)$. In our simulations, we set $n = 10^3$ and $k \in \lbrace 13,14 \rbrace$ for different values of $p$ and show that the transitive closure step can significantly improve the simulated output of the $k$-core estimator for both ER and non-ER models.

In the attached PDF, the $k$-core estimator is simulated to obtain a partial matching between any two graphs $G_i$ and $G_j$. The fraction of correctly matched nodes for each pairwise matching is tabulated for the example case of $m=6$. Next, each pairwise matching is boosted using transitive closure, and the fraction of correctly matched nodes after boosting is tabulated alongside. By averaging these fractions over the $\binom{m}{2}$ pairwise matchings, we obtain the mean fraction of correctly matched nodes before and after transitive closure. This is plotted against $m$ in the accompanying figures.

The simulations consider three scenarios:
1. Erdos-Renyi graphs with low correlation, $s=0.25$: Each graph provides a lot of new information in this setting, and so the fraction of correctly matched nodes among any pair of graphs is significantly boosted with transitive closure.
2. Erdos-Renyi graphs with high correlation, $s=0.8$. Since the graphs are all similar to each other, the boosting is less prominent compared to the low correlation scenario.
3. Non Erdos-Renyi graphs (stochastic block model, SBM). To demonstrate the versatility of the transitive closure step, we simulate correlated SBMs. The correlated SBMs are obtained from the subsampling model where the parent graph is an SBM with 5 equal-sized clusters. Within each cluster, edges are independently present with probability $p$, whereas they are independently present with probability $q$ between clusters. Here too, there is a significant boost in the mean fraction of correctly matched nodes (see Table 3 and Figure 3). This suggests that transitive closure can be a useful bridge between $2$-ary and $m$-ary graph matching for more general models in practice.

---

### Comment · Area_Chair_XKFJ · 2024-08-08

Dear authors, dear reviewers,

the discussion period has begun as the authors have provided their rebuttals.
I encourage the reviewers to read all the reviews and the corresponding rebuttals: the current period might be an opportunity for further clarification on the paper results and in general to engage in an open and constructive exchange.

Many thanks for your work.
The AC

---

### Decision · Program_Chairs · 2024-09-25

**Decision:**

Reject

**Comment:**

The paper considers the problem of graph alignment given a set of $m$ correlated Erdos-Renyi random graphs. An exact criterion is proven to characterize the boundary of the feasibility region of the alignment in parameter space. Two algorithms are proposed: both are shown to reach the information-theoretical threshold for recovery.

The submission is an interesting contribution to the theory of graph alignment within the theoretical framework of correlated Erdos-Renyii graphs and it has been indeed appreciated by the Referees under this point of view: the extension from the $m=2$ to the $m>2$ case is nontrivial and might boost further investigation into the benefits of using multiple correlated graphs in alignment problems.

On the other hand, the main concern about the submission is related to its limitation to the very specific graph ensemble and the lack of analysis of the computational aspects in terms of numerical tests. More importantly, the contribution appears to be focused more on a genuine probability and graph theoretical problem than on the theory of learning and its computational aspects. Although works in this spirit have been accepted in NeurIPS in the past, after a discussion with the Senior Area Chair we concurred on the fact that such previous submissions were also to be intended out of the scope of the program.  As a result, we regret to have to reject the submission and we recommend a better suited conference or journal where the paper could likely reach a more appropriate and engaged audience.